# Personalized Policy Learning through Discrete Experimentation

**Zhiqi Zhang** [1]  **Zhiyu Zeng** [2]  **Ruohan Zhan** [3]  **Dennis Zhang** [1]

## Abstract

While Randomized controlled trials (RCTs), or A/B tests, are the gold standard for optimizing online-platform policies, they are limited by discrete testing levels. This approach is suboptimal for continuous variables (e.g., prices and incentives), as it fails to extrapolate to untested values or account for user heterogeneity. We address this by developing Deep Learning for Policy Targeting (DLPT) to learn personalized continuous policies from discrete RCTs using high-dimensional features. We prove our estimators are asymptotically unbiased and consistent, achieving a $\sqrt{n}$-regret bound. In a collaboration with a leading social media platform to optimize creator incentives, we show that DLPT substantially outperforms existing benchmarks.

## 1. Introduction

Randomized Controlled Trials (RCTs) are the gold standard for policy evaluation in online systems (Kohavi & Thomke, 2017). While effective for discrete choices, standard RCT protocols face a structural limitation when optimizing continuous decision variables, such as pricing or incentives. Due to sample complexity constraints, continuous action spaces are typically discretized into a finite set of treatment arms (Feldman et al., 2022; Zhang et al., 2020). Consequently, current industry practice involves selecting the best-performing discretized level (Christensen & Osman, 2023; Cohen et al., 2016). This approach is inherently suboptimal: it precludes extrapolation to the full continuous manifold and fails to capture complex interactions between treatments and high-dimensional user heterogeneity, resulting in coarse-grained, non-personalized policies.

[1]Olin Business School, Washington University in St. Louis, St. Louis, Missouri, US [2]Antai College of Economics and Management, Shanghai Jiao Tong University, Shanghai, China [3]School of Management, University College London, London, UK. Correspondence to: Zhiqi Zhang <z.zhiqi@wustl.edu>.

*Proceedings of the 43$^{rd}$ International Conference on Machine Learning*, Seoul, South Korea. PMLR 306, 2026. Copyright 2026 by the author(s).

In this paper, we aim to address the following research questions. First, how can we accurately estimate policy value over a continuous treatment space, conditioned on high-dimensional features, when observing only a discrete number of experimental arms on this continuous variable? Furthermore, how can we derive the optimal personalized policy for each feature combination? Our goal is to develop a theoretical framework with provable guarantees and rigorously validate our methods through real-world RCTs and applications.

Theoretically, we propose a Deep Learning framework for Policy Targeting (DLPT). Our model assumes a semi-parametric data generation process, where outcomes depend parametrically on continuous treatments but non-parametrically on high-dimensional context features. We implement a three-stage estimation procedure: (1) estimating nuisance parameters via neural networks, (2) estimating policy values using universal orthogonal score functions to ensure asymptotic unbiasedness and semi-parametric efficiency, and (3) optimizing these values to learn the personalized policy. We prove that our learned policy achieves a regret bound of $O\left(r\left(1+\sqrt{\log(1/r)}\right)\sqrt{d/n}+r\sqrt{\log(1/\delta)/n}\right)$ where $r$ specifies the distance between policies in class $\Pi$ and $d$ represents complexity. Notably, our results demonstrate that DLPT achieves the minimax optimal regret rate of $\sqrt{n}$ within a fixed policy class.

Empirically, we validate our framework through a large-scale field experiment with "Platform O," a leading social media platform. The RCT, involving 12.3% of the user base ($N \approx 7.4$ millions), tested discrete monetary incentive levels to encourage user-generated content. We assess performance on two fronts: estimation accuracy and policy optimization. For out-of-sample Average Treatment Effect (ATE) recovery, DLPT achieves a Mean Absolute Percentage Error (MAPE) of 2.25%, significantly outperforming benchmarks (7.18% to 43.59%). In policy learning, evaluated on held-out user subgroups, DLPT attains the lowest Mean Percentage Regret (MPR) of 2.22%, compared to 6.32% to 10.05% for baseline methods.

Our study makes several pivotal contributions to both academic research and practical applications in experimentation and platform analytics. From a methodological perspective,

we are the first, to our best knowledge, to propose a framework for estimating policy values and learning personalized policy over a continuous treatment variable when only observing discrete sets of treatment levels. The proposed framework and estimation procedure guarantee a $\sqrt{n}$ regret bound for the learned policy. In real-world settings, we have validated our method in collaboration with a major online platform. The results confirm that our framework not only upholds strong theoretical promises but also excels in practical deployments, effectively handling extensive user features and nuanced treatment effects. More broadly, we develop our method for post-experiment analysis, allowing for seamless integration into existing experimental platforms. This facilitates its adoption by practitioners who can readily apply our approach to enhance decision-making processes without modifying their existing experimentation infrastructure.

### Conflict of Interest Disclosure

The authors Z.Z. and Z.Z. were unpaid interns at the collaborating company ("Platform O") within the three years prior to the submission of this work.

## 2. Literature Review

Our research builds on three key streams of literature: personalized pricing, policy learning and causal Machine Learning (ML), and experimentation on platforms.

**Personalized Pricing**. Empirical demand estimation has traditionally relied on parametric models, such as MNL and random coefficient models (Rossi et al., 1996; Smith et al., 2023; Aryal et al., 2024), or nonparametric approaches (Hausman & Newey, 1995; Chen & Gallego, 2021). Recent work integrates machine learning to handle high-dimensional data (Bajari et al., 2015; Qi et al., 2022; Dubé & Misra, 2023). While methodologies like IPW and CATE (Hitsch et al., 2024) identify optimal policies, they typically select from discrete candidate sets or require observational data with wide support. Table 5 in the Appendix summarizes representative literature in applied research of personalized pricing. We contribute by proposing a double machine learning (DML) framework that enables causal inference and personalization over a *continuous* policy space, even when trained on data from discrete-arm experiments.

**Policy Learning and Causal ML**. Offline policy learning is often categorized by the action space. For discrete actions, recent advances focus on doubly robust scoring (Athey & Wager, 2021), multi-action learning (Zhou et al., 2023), and adaptive settings (Zhan et al., 2024). For continuous actions, existing estimators often require randomized trials with continuous logging propensities (Chen et al., 2016) or rely on kernel smoothing for local evaluation (Kallus & Zhou, 2018). In our setting, the treatment support consists of exactly five discrete arms from a randomized experiment. Applying kernel-based continuous estimators in this regime would require strong interpolation assumptions across wide gaps in the treatment space. Our DLPT framework addresses this challenge through a structured polynomial model, acting as an adaptation of these continuous methods to the sparse discrete-arm RCT setting. Our work builds on the semiparametric efficiency framework of (Chernozhukov et al., 2019) and the DML theory of (Chernozhukov et al., 2018). While DML has been successfully applied to nuisance parameter estimation via deep neural networks (Farrell et al., 2021; Knaus, 2022; Ye et al., 2023), we extend this framework specifically to overcome this structural mismatch, providing a rigorous implementation (DLPT) that extrapolates optimal treatments with provable $\sqrt{n}$-regret bounds.

**Experimentation on Platforms**.Research on platform experimentation largely focuses on experimental design, addressing challenges like interference (Wager & Xu, 2021; Holtz et al., 2024), adaptive sequential assignment (Gur & Momeni, 2022; Zhao & Zhou, 2024), and two-sided market bias (Johari et al., 2022). In contrast, our work targets *post-hoc* analysis. While post-hoc methods exist for discrete causal inference (Yu et al., 2022; Zeng et al., 2023), our framework enhances operational flexibility by enabling continuous policy optimization from standard A/B tests. This allows platforms to extract value from existing experimental infrastructure without requiring complex, multi-arm, or continuous-randomization designs.

## 3. Theoretical Framework

In this section, we introduce our theoretical framework, DLPT, for personalized policy learning. The framework first extrapolates the causal effects of discrete experimented treatments to the entire continuous space, and then finds the optimal personalized continuous-treatment policy based on the extrapolation. We start with formalizing the problem and then propose our method accompanied with its theoretical guarantees.

**Problem Setup.**[1] A platform conducts a randomized experiment to determine the optimal treatment level within a continuous treatment space $\mathcal{T} = [0, \bar{T}] \subset \mathbb{R}$, where $\bar{T} < \infty$, with the goal of maximizing an outcome of interest (e.g., revenue). Due to operational constraints, the platform can only implement $m$ discrete treatment levels, $\{t_1, t_2, \ldots, t_m\} \subset \mathcal{T}$, during the experiment. Let $T$ denote the treatment level assigned to a user. The platform observes the assigned treat-

---

[1] *On notations:* Throughout the paper, vectors and matrices are denoted in boldface. Vectors are treated as column vectors, and $\boldsymbol{v}'$ denotes the transpose of vector $\boldsymbol{v}$. Random variables are represented by capital letters, and their realizations by lowercase letters.

ment level $T$, as well as high-dimensional and bounded pretreatment covariates $\boldsymbol{X} \in \mathcal{X} \in \mathbb{R}^{d_{\boldsymbol{X}}}$. Conditional on covariates, each treatment level is assigned with equal probability,[2] i.e., $\mathbb{P}(T = t_j \mid \boldsymbol{X} = \boldsymbol{x}) = \frac{1}{m}, \quad \forall j \in \{1, \ldots, m\}$. After treatment assignment, the user reveals an outcome $Y \in \mathcal{Y} \subset \mathbb{R}$, which the platform can observe. Without loss of generality, we focus on binary outcomes $Y \in \{0, 1\}$; our results extend directly to general bounded outcomes.

Following (Farrell et al., 2020), we assume that the underlying data generation process (DGP) has a semi-parametric specification such that

$$\mathbb{E}[Y | \boldsymbol{X} = \boldsymbol{x}, T = t] = G(\boldsymbol{\theta}^*(\boldsymbol{x}), t). \tag{1}$$

Above, the link function $G(\cdot)$ is parametrically specified a priori, which can vary from linear to non-linear forms, such as sigmoid, depending on the specific domain knowledge and hypotheses stipulated by the researchers. The nonparametric part, $\boldsymbol{\theta}^*(\cdot)$, is typically known as "nuisance parameter" in causal inference literature (Newey, 1994), which is designed to be highly adaptable—without parametric restrictions—to encapsulate a summary of user characteristics and their interactions with the treatment level $t$. This $\boldsymbol{\theta}^*(\cdot)$ shall be learned using deep neural networks (DNNs) in our empirical settings, which afford the flexibility needed to capture complex, non-linear relationships that may exist between user attributes and treatment effects.

In our analysis, we assume a specific sigmoid-polynomial specification for $G(\cdot)$ as the DGP. Accordingly, specification (1) is detailed as follows:

$$G(\boldsymbol{\theta}_K^*(\boldsymbol{x}), t) = \frac{1}{1 + \exp(-(\boldsymbol{\theta}_K^*(\boldsymbol{x})' \tilde{\boldsymbol{T}}_K))}, \tag{2}$$

where $\tilde{\boldsymbol{T}}_K(t) = (1, t, t^2, \ldots, t^K)' \in \mathbb{R}^{K+1}$ denotes the degree-$K$ polynomial feature vector constructed from the treatment variable $t$. The coefficient vector $\boldsymbol{\theta}_K^*(\boldsymbol{x}) \in \mathbb{R}^{K+1}$ varies with covariates $\boldsymbol{x}$, and governs the influence of each polynomial term in the transformed treatment input. As to be discussed in Remark 3.2, the optimal $K$ is chosen as the number of discrete treatments minus 1.

**Policy.** A policy $\pi$ is a treatment assignment rule that maps from the covariate space $\mathcal{X}$ to the treatment space $\mathcal{T}$. For a given user with covariates $\boldsymbol{x}$, the value $\pi(\boldsymbol{x})$ denotes the treatment level that the policy prescribes for that user based on her characteristics. Let $H(\cdot)$ denote the value function chosen by researchers or practitioners. For example, $H(\boldsymbol{\theta}_K^*(\boldsymbol{x}), \pi(\boldsymbol{x})) = wG(\boldsymbol{\theta}_K^*(\boldsymbol{x}), \pi(\boldsymbol{x})) - c\pi(\boldsymbol{x})$ represents net benefit as the profit gained from users outcome with weight $w$ minus policy cost with unit cost $c$. Here, $w$ is the unit profit from user engagement (a video upload), derived

from downstream value like ad revenue and retention. $c$ is the exact unit cost of the incentive (e.g., 100 Points $\approx 0.01$ USD). For a given policy $\pi$, we use $V_K(\pi)$ to denote its policy value, which represents the expected value when the treatment assignment follows policy $\pi$:

$$V_K(\pi) := \mathbb{E}[H(\boldsymbol{\theta}_K^*(\boldsymbol{x}), \pi(\boldsymbol{x}))], \tag{3}$$

where the expectation is taken over the covariate distribution.

**Goal.** Given a pre-specified policy class $\Pi$ (such as linear policies or decision-tree policies) and the current semi-parametric DGP assumption $G(\boldsymbol{\theta}_K^*(\boldsymbol{x}), t)$, we aim to learn a policy $\hat{\pi} \in \Pi$ from the data, such that its policy value is maximized, or equivalently, we aim to minimize its regret $R_K(\hat{\pi})$, which measures the policy's suboptimality as defined below:

$$R_K(\hat{\pi}) = V_K(\pi_K^*) - V_K(\hat{\pi}), \tag{4}$$

where $\pi_K^* = \arg\max_{\pi \in \Pi} V_K(\pi)$ denotes the optimal *in-class* policy.

**Three-Stage Policy Learning Method:** DLPT. We propose a three-stage method to learn optimal policies, which we term as Deep Learning Policy Targeting (DLPT). The key is to causally extrapolate the policy value of continuous treatment $T$ from discrete observed experiments. Our method unfolds in the following stages: (i) estimating the nuisance parameter $\boldsymbol{\theta}_K^*(\boldsymbol{x})$ via structured ML models; (ii) estimating counterfactual policy values using Neyman orthogonal scores; and (iii) learning the optimal personalized policy within the specified policy class. We detail each stage in sections 3.1, 3.2, and 3.3 respectively and complement these procedures with theoretical guarantees for policy value estimation and policy learning.

In later sections, we shall instantiate our DLPT method in the field. Specifically, our empirical setting considers outcome $Y$ as a binary variable indicating user video uploads, the treatment $T$ as a continuous variable denoting monetary incentive offered to encourage user creation, and $\boldsymbol{X}$ as users' features (See Section C in Appendix for detailed empirical setting). Our goal is to learn a cost-effective policy for offering monetary incentives. Although this illustration is specific to our empirical application, our DLPT method is designed to be flexible and can be customized to accommodate various continuous outcome variables. Let $n$ denote the number of users involved in the experiment, indexed by $i \in \{1, \ldots, n\}$. For each user $i$, let $\boldsymbol{X}_i$, $T_i$, and $Y_i$ denote her covariates, treatment level, and observed outcome, respectively. At the conclusion of the experiment, the platform collects $n$ i.i.d. observations: $\{\boldsymbol{Z}_i = (\boldsymbol{X}_i', T_i, Y_i)'\}_{i=1}^n$.

---

[2] Our method does not require uniform treatment assignment; this assumption is made purely for clarity of exposition.

### 3.1. Stage 1: Nuisance Parameter Estimation

In the first stage, we approximate the link function $G(\cdot)$ using a structured DNN. Specifically, we employ a flexible feed-forward DNN, denoted by $\hat{\boldsymbol{\theta}}(\boldsymbol{x})$, to approximate the nuisance parameter $\boldsymbol{\theta}_K^*(\boldsymbol{x})$. Following our sigmoid-polynomial specification introduced in Equation (2), the interaction between the treatment variable $t$ and the estimated covariate-dependent coefficients $\hat{\boldsymbol{\theta}}(\boldsymbol{x})$ is modeled in a structured and explicit form:

$$G(\hat{\boldsymbol{\theta}}(\boldsymbol{x}), t) \tag{5}$$

$$= \frac{1}{1 + \exp\left(-\left(\hat{\theta}_0(\boldsymbol{x}) + \hat{\theta}_1(\boldsymbol{x})t + \cdots + \hat{\theta}_K(\boldsymbol{x})t^K\right)\right)}, \tag{6}$$

where $\hat{\theta}_k(\boldsymbol{x})$ denotes the coefficient corresponding to the $k$-th polynomial term of the treatment variable, for $k = 0, \ldots, K$, as a function of user covariates $\boldsymbol{x}$. The nuisance parameter estimator $\hat{\boldsymbol{\theta}}(\cdot)$ is learned by solving the following empirical risk minimization (ERM) problem:

$$\hat{\boldsymbol{\theta}}(\cdot) = \arg\min_{\tilde{\boldsymbol{\theta}} \in \boldsymbol{\Theta}} \frac{1}{n} \sum_{i=1}^{n} \ell(Y_i, T_i, \tilde{\boldsymbol{\theta}}(\boldsymbol{X}_i)), \tag{7}$$

where $\ell(\cdot)$ is an appropriate loss function (e.g., squared loss for continuous outcomes or cross-entropy loss for binary outcomes), and $\boldsymbol{\Theta}$ denotes the functional space of all possible feed-forward DNN under a chosen set of hyperparameters.

To achieve a sufficiently rapid convergence rate of the nuisance estimates—which is crucial for Stage 2 (policy value estimation) and Stage 3 (policy learning)—we establish certain regularity conditions. These conditions are commonly assumed in the literature with feed-forward neural netwotk (Farrell et al., 2020), pertain to the continuity and curvature of loss function $l(\cdot)$ (Assumption B.1 in Appendix B.1), the joint distribution of $(\boldsymbol{X}', T, Y)'$ (Assumption B.2(a) in Appendix B.1), and DGP nuisance parameter $\boldsymbol{\theta}^*(\boldsymbol{x})$ (Assumption B.2(b) in Appendix B.1). Given the structure of $\tilde{\boldsymbol{T}}_K(t) = (1, t, t^2, \ldots, t^K)'$, we analogously define the policy-induced polynomial vector as $\tilde{\boldsymbol{\pi}}_K(\boldsymbol{x}) = (1, \pi(\boldsymbol{x}), \pi(\boldsymbol{x})^2, \ldots, \pi(\boldsymbol{x})^K)'$. Proposition 3.1 presents the convergence rate for estimating nuisance parameters during Stage 1, with its proof deferred to Appendix B.2. This result is foundational to ensuring the reliability and efficiency of subsequent stages of DLPT implementation.

**Proposition 3.1** (IDENTIFIABILITY AND CONVERGENCE OF NUISANCE ESTIMATE). *Suppose that $\mathbb{E}[\tilde{\boldsymbol{T}}_K \tilde{\boldsymbol{T}}_K' | \boldsymbol{X}]$ is positive definite across all $\boldsymbol{X}$, and that Assumption B.1 and Assumption B.2 in Appendix B.1 hold,*

*(a) The nuisance parameter function $\boldsymbol{\theta}_K^*(\boldsymbol{x})$ can be non-parametrically identified in DGP (5).*

*(b) (Farrell et al., 2020; 2021) If the structured DNN has width $H = O(n^{d_C/2(p+d_C)} \log^2 n)$ and depth $L = O(\log n)$, there exists a positive constant $C$ that depends on the fixed quantities in Assumption B.2, such that with probability at least $1 - \exp(n^{-d_C/(p+d_C)} \log^8 n)$, for $n$ large enough, the convergence of $\hat{\boldsymbol{\theta}}$ satisfies*

$$\mathbb{E}[\|\hat{\boldsymbol{\theta}}(\boldsymbol{x}) - \boldsymbol{\theta}_K^*(\boldsymbol{x})\|_{L_2(\boldsymbol{X})}^2] \tag{8}$$

$$= O\left(n^{-\frac{p}{p+d_C}} \log^8(n) + \frac{\log \log n}{n}\right), \tag{9}$$

*where $p$ characterizes the smoothness of $\boldsymbol{\theta}_K^*(\cdot)$, and $d_C$ is the dimension of continuous features in context.*

*Remark* 3.2 (CHOOSING OPTIMAL $K$). The uniform positive definiteness condition for $\mathbb{E}[\tilde{\boldsymbol{T}}_K \tilde{\boldsymbol{T}}_K' | \boldsymbol{X}]$ is satisfied when each user is randomly assigned $m \geq K + 1$ discrete treatment values, regardless of their covariates $\boldsymbol{X}$. The higher $K$ in Equation (5), the better the approximation power our method attains. Therefore, if an experiment with $m$ treatment conditions is run, then the optimal $K$ should be $m - 1$ when applying our method.

### 3.2. Stage 2: Policy Value Estimation

In the second stage of our framework, we estimate the policy value of each counterfactual policy $\pi$. Recall that the policy value is defined as:

$$V_K(\pi) := \mathbb{E}[H(\boldsymbol{\theta}_K^*(\boldsymbol{x}), \pi(\boldsymbol{x}))] \tag{10}$$

$$= \mathbb{E}[wG(\boldsymbol{\theta}_K^*(\boldsymbol{x}), \pi(\boldsymbol{x})) - c\pi(\boldsymbol{x})], \tag{11}$$

where $w$ represents the unit profit gained from users' engagement outcomes, and $c$ denotes the unit cost of the policy. This formulation captures the expected net benefit of implementing policy $\pi$.

With the nuisance estimate $\hat{\boldsymbol{\theta}}(\cdot)$ obtained from Stage 1, a natural policy value estimator is to plug $\hat{\boldsymbol{\theta}}(\cdot)$ into Equation (10). However, the estimation error of $\hat{\boldsymbol{\theta}}(\cdot)$ will propagate to the policy value estimation, and the error diminishing rate guaranteed in Proposition 3.1 will be not fast enough to guarantee valid inference and prepare for efficient policy learning.

In light of this, we follow (Farrell et al., 2020) and propose a semi-parametric policy value estimator by using orthogonalized score functions of $\hat{\boldsymbol{\theta}}(\cdot)$. Score functions with this orthogonality condition, or more precisely, universal (Neyman) orthogonality condition, are typically referred to as influence functions in econometrics literature (Newey, 1994; Ichimura & Newey, 2022). These influence functions exhibit first-order robustness to small variations in nuisance parameter estimation, by having error decay rate faster than $\sqrt{n}$. This robustness allows for semi-parametrically inference and efficient policy learning, even if the convergence

speed of the nuisance estimates is not sufficient to achieve semi-parametric efficiency itself.

**Proposition 3.3** (INFLUENCE FUNCTION). *Assume Assumptions B.1, B.2, and B.3 in Appendix B.1 hold. $\psi(z, \pi; \hat{\theta}, \Lambda) - V_K(\pi)$ is an influence function of $V_K(\pi)$, where $\psi(z, \pi; \hat{\theta}, \Lambda)$ is referred to as score function and is defined as:*

$$\psi(z, \pi; \hat{\theta}, \Lambda) = H(\hat{\theta}(x), \pi(x)) \tag{12}$$
$$- H_\theta(\hat{\theta}(x), \pi(x))\Lambda(x)^{-1}\ell_\theta(y, t, \hat{\theta}(x)), \tag{13}$$

*where $z = (x', t, y)'$ is the observed data sample, $\ell_\theta$ is the gradient of $\ell$ with respect to $\theta$, $\Lambda(x) = \mathbb{E}[\ell_{\theta\theta}(y, t, \theta(x))|X = x]$ as the expectation of second order derivative of $\ell$ with respect to $\theta$, and $H_\theta(\hat{\theta}(x), \pi(x))$ is the gradient of $H$ with respect to $\theta$.*

*Remark* 3.4 (Explicit computation of nuisance parameter $\Lambda(x)$). Unlike general scenarios explored in (Farrell et al., 2020), our empirical setting allows for a direct computation of the nuisance parameter $\Lambda(x)$, obviating the need for its estimation. This simplification is possible because the distribution of the treatment level $t$, defined by the random treatment assignment mechanism, is known a priori. Depending on the loss function employed in Stage 1 of our analysis, the form of $\Lambda(x)$ varies as follows:

$$\Lambda(x) = \begin{cases} \mathbb{E}\left[G_\theta(\hat{\theta}(x), t)G_\theta(\hat{\theta}(x), t)' \mid X = x\right], \\ \text{if using MSE loss} \\ \mathbb{E}\left[G_\theta(\hat{\theta}(x), t)\tilde{T}_K' \mid X = x\right], \\ \text{if using Binary Cross Entropy loss} \end{cases}$$

where $G_\theta(\hat{\theta}(x), t) = G(\hat{\theta}(x), t)(1 - G(\hat{\theta}(x), t))\tilde{T}_K$.

With the score function in (12), the below result shows that its derivative with respect to our nuisance parameters evaluated at the true parameter values is 0. In other words, it satisfies Neyman orthogonality.

**Proposition 3.5** (UNIVERSAL ORTHOGONALITY). *The score function $\psi(z, \pi; \hat{\theta}, \Lambda)$ defined in Proposition 3.3 for estimating policy value $V_K(\pi)$ is universal orthogonal with respect to the nuisance parameter $\theta$. That is,*

$$\mathbb{E}[\nabla_\theta \psi(z, \pi; \theta^*, \Lambda)|X = x] = 0. \tag{14}$$

Now we are ready to introduce our policy value estimator $\hat{V}_K^{\text{DLPT}}(\pi)$, based on the score function in Proposition 3.3, composed with the sample-splitting estimation procedure. Specifically, we randomly partition the dataset $S$, consisting of $n$ i.i.d. samples, evenly into two subsets $S_1$ and $S_2$. We

first estimate the nuisance parameters $\hat{\theta}$ using subset $S_1$ following the procedure outlined in Stage 1. Subsequently, we utilize subset $S_2$ for policy evaluation in Stage 2.

$$\hat{V}_K^{\text{DLPT}}(\pi) = \frac{1}{|S_2|} \sum_{i \in S_2} \psi(Z_i, \pi; \hat{\theta}, \Lambda) \tag{15}$$

$$\hat{\Psi}_K^{\text{DLPT}}(\pi) = \frac{1}{|S_2|} \sum_{i \in S_2} \left(\psi(Z_i, \pi; \hat{\theta}, \Lambda) - \hat{V}_K^{\text{DLPT}}(\pi)\right)^2 \tag{16}$$

Proposition 3.6, adapted from Theorem 3 in (Farrell et al., 2020), presents the $\sqrt{n}$-asymptotic normality of our policy value estimator $\hat{V}_K^{\text{DLPT}}(\pi)$ with the estimated variance $\hat{\Psi}_K^{\text{DLPT}}(\pi)$, allowing for policy value inference and hypothesis testing. It also shows that, with the Neyman orthogonality score function, our estimator is consistent and asymptotically normal.

**Proposition 3.6** (ASYMPTOTIC NORMALITY). *Assume Assumptions B.1, B.2, and B.3 in Appendix B.1 hold, and $\Lambda(x)$ is uniformly invertible. Also, the nuisance parameter estimator $\hat{\theta}_1$ in Stage 1 obeys $\|\hat{\theta}_1 - \theta_1^*\|_{L_2(X)} = o(n^{-1/4})$, which is verified by Proposition 3.1(b). Then*

$$\sqrt{n}\hat{\Psi}_K^{\text{DLPT}}(\pi)^{-1/2}(\hat{V}_K^{\text{DLPT}}(\pi) - V_K(\pi)) \to_d \mathcal{N}(0, 1). \tag{17}$$

Proposition 3.6 shows that we can construct an semi-parametric estimator for evaluating any personalized policy $\pi$ from the data and conduct valid inference. Note that this estimator can be used to evaluate any policies, not limited to the observed levels in the experiment. With this estimator, we next discuss how to learn the optimal personalized policy from the data.

### 3.3. Stage 3: Policy Learning

We denote the optimal policy derived from our framework as $\hat{\pi}_{DLPT}^\Pi$, defined as the policy within a specified class $\Pi$ that maximizes the expected policy value. Formally, this policy $\hat{\pi}_{DLPT}^\Pi$ is obtained by solving the constrained empirical risk minimization (ERM) problem:

$$\hat{\pi}_{DLPT}^\Pi = \inf_{\pi \in \Pi} \left\{-\hat{V}_K^{\text{DLPT}}(\pi)\right\} \tag{18}$$
$$= \inf_{\pi \in \Pi} \left\{-\mathbb{E}_{S_2}[\psi(z_i, \pi; \hat{\theta}, \Lambda)]\right\}, \tag{19}$$

where $\mathbb{E}_{S_2}[\cdot]$ denotes the empirical average over sample $S_2$. The result below provides the regret guarantee of the learned policy.

**Theorem 3.7.** *Define the function class: $\psi \circ \Pi = \{\psi(\cdot, \pi; \hat{\theta}, \Lambda) : \pi \in \Pi\}$. Let $d$ be the VC subgraph dimension (also known as pseudo dimension) of $\psi \circ \Pi$. Let $r = \sup_{\pi \in \Pi} \|\psi(\cdot, \pi; \hat{\theta}, \Lambda) - \psi(\cdot, \pi^*; \hat{\theta}, \Lambda)\|_{L_2(X)}$. Assume Assumptions B.1, B.2, and B.3 in Appendix B.1 hold,*

*then with probability $1 - \delta$,*

$$R_K(\hat{\pi}_{DLPT}^{\Pi}) = O\left( r\left(1 + \sqrt{\log(1/r)}\right)\sqrt{\frac{d}{n}} + r\sqrt{\frac{\log(1/\delta)}{n}} \right). \tag{20}$$

Theorem 3.7 shows that the learned policy achieves minimax optimality, with regret scaling at the rate $1/\sqrt{n}$. This rate is consistent with results in policy learning literature for discrete treatment (Athey & Wager, 2021; Zhou et al., 2023). In other words, our model permits far more flexible functional forms and continuous optimization variables, yet its policy-learning regret remains as efficient as in the original discrete-experiment setting. The proof of Theorem 3.7 is provided in Appendix B.5.

*Remark* 3.8 (Role of policy class). The regret bound in Theorem 3.7 explicitly accounts for policy class complexity, measured by VC subgraph dimension. This quantity captures a fundamental trade-off in policy learning: more complex policies are harder to learn effectively. However, characterizing the VC subgraph dimension can be challenging—it may even be infinite for some parametric function classes with finite-dimensional parameters (Shalev-Shwartz & Ben-David, 2014). As an alternative, Appendix B.5 presents a slightly more involved regret bound in Proposition B.8, expressed in terms of covering numbers. Covering number is another standard measure of function class complexity and can be upper bounded by the VC subgraph dimension (Van der Vaart, 2000). Note that covering numbers are often more tractable to analyze and have been studied for many common function classes, including those satisfying Lipschitz conditions (Vershynin, 2010; Xu & Zeevi, 2020).

In practice, achieving this regret bound requires controlling this complexity to prevent overfitting. When the continuous policy is parameterized by a Deep Neural Network (DNN), the policy class complexity is implicitly regularized by the network architecture (e.g., width $W$ and depth $L$) and training techniques such as early stopping. Specifically, the covering number $\log \mathcal{N}_2(\epsilon, \Pi, n)$ for a ReLU DNN scales as $O(W^2 L \log(WL/\epsilon))$, meaning the statistical complexity and the resulting regret bound are controlled by the finite network architecture rather than the ambient continuous treatment space.

## 4. Empirical Setting and Data

To validate DLPT, we partnered with "Platform O," a leading global short-video platform, to conduct a large-scale randomized field experiment. The platform sought to optimize financial incentives to encourage content creation among low-activity users. This setting provides an ideal testbed for our framework: while the decision variable (monetary reward) is conceptually continuous, the platform could only test a sparse set of discrete treatment levels due to sample

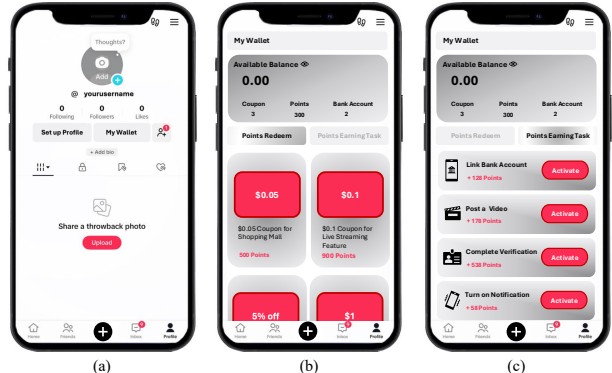

Figure 1: Illustration of the 'Point" System on Platform

Note: To protect Platform O's identity and obscure nonessential details, we have slightly modified the interface of a widely used video-sharing platform.

size and operational constraints. Appendix C provides the details.

### 4.1. Experiment Design and Data Collection

The experiment ran from August 23 to 29, 2021, targeting $N = 7,349,648$ low-activity creators (defined as users posting videos on fewer than three days in the preceding month). Users were randomly assigned to a control group or one of four treatment groups.

**Treatments** ($T$). The treatment variable $T_i \in \{0, 178, 358, 538, 718\}$ represents the "Points" offered to user $i$ for completing a "Post a Video" task (100 Points $\approx$ 0.01 USD). Control users ($T_i = 0$) did not receive the offer (See Figure 1).

**Outcome** ($Y$). The outcome $Y_i \in \{0, 1\}$ is a binary variable indicating whether user $i$ posted at least one video during the experimental period. As detailed in Appendix C, this binary definition aligns with the platform's strategic focus on habit formation rather than volume.

**Covariates** ($X$). We observe high-dimensional pre-treatment covariates $\boldsymbol{X}_i \in \mathbb{R}^{77}$ (after one-hot encoding), derived from 12 categorical variables (e.g., gender, location) and 46 continuous variables (e.g., historical upload frequency).

### 4.2. Model-Free Evidence

Before applying DLPT, we verify two prerequisites for personalized continuous policy learning: (1) the treatment significantly affects the outcome, and (2) there is heterogeneity in user response.

As detailed in Appendix D, OLS regressions confirm a monotonic positive relationship between reward size and upload probability, with relative effect sizes ranging from

13.01% to 19.69% ($p < 0.0001$). Crucially, we observe diminishing marginal returns at higher reward levels (See Table 1). Furthermore, we detect significant heterogeneity: users with lower historical engagement are less responsive to incentives, while heavy content consumers are more responsive (interaction $p$-values $< 0.01$). This heterogeneity and non-linearity confirm the potential value of optimizing a personalized, continuous reward function.

Table 1: Average Treatment Effects of Point Rewards

| | Dependent Variable: Video Uploading ($Y$) | | | |
|---|---|---|---|---|
| | 178 | 358 | 538 | 718 |
| Treatment | 0.0586**** | 0.0729**** | 0.0824**** | 0.0886**** |
| (Std. Error) | (0.0011) | (0.0011) | (0.0011) | (0.0011) |
| Relative Effect | 13.01% | 16.20% | 18.30% | 19.69% |
| Observations | 2,957,138 | 2,957,749 | 2,957,291 | 2,957,886 |

Note: To preserve data confidentiality, we normalize the dependent variable. Robust standard errors are reported in parentheses. ****$p < 0.0001$.

# 5. Evaluation with the Field Data

In this section, we empirically evaluate the DLPT framework using the field experiment data described in Section 4. We employ a cross-evaluation strategy: we train the model on a subset of users and test its ability to (1) recover the ground-truth Average Treatment Effect (ATE) for withheld treatment arms, and (2) identify optimal policies for withheld user subgroups. Details of the implementation can be found in Appendix E.

## 5.1. Implementation and Benchmarks

**Implementation.** We follow the three-stage procedure outlined in Section 3. First, we model the data generating process using a structured Deep Neural Network (DNN) with polynomial degree $K = 3$ (See Figure 2). Second, we construct doubly robust estimators for the ATE and policy value using Neyman Orthogonal score functions on an inference set $\mathcal{S}_{\text{inference}}$. Third, we solve the Empirical Risk Minimization (ERM) problem to identify the optimal personalized policy.

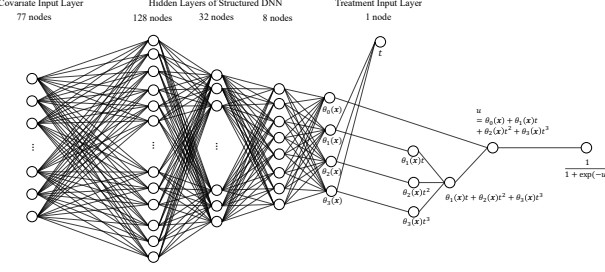

Figure 2: Structured DNN Architecture for Nuisance Estimation

**Benchmarks.** We compare DLPT against five established

baselines: Linear Regression (LR), Logistic Regression (LogR), Logit Choice Model (LCM), Pure Deep Learning (PDL), Generalized Random Forest (GRF) T-learner, non-personalized Cubic Spline (Spline), and a non-personalized Uniform Policy (UNI). All benchmarks are trained and evaluated on identical data splits. Detailed descriptions and implementations can be found in Appendix E.

## 5.2. ATE Recovery Results

We first evaluate the estimation accuracy of our framework by assessing its ability to recover Average Treatment Effects (ATEs) for held-out treatment levels. This cross-validation exercise tests whether DLPT can accurately extrapolate treatment effects to levels not used during training.

For each of the four treatment levels $t \in \{178, 358, 538, 718\}$, we implement the following procedure: (1) mask all outcomes for users assigned to treatment level $t$, (2) train each method on the remaining three treatment levels plus the control group, (3) use the trained model to predict the ATE at the held-out level $t$, and (4) compare the predicted ATE to the ground truth. We measure accuracy using Mean Absolute Percentage Error (MAPE) and Mean Squared Error (MSE), both averaged across the four held-out levels.

Table 2 reports the results. DLPT achieves the lowest MAPE of 2.25% and MSE of 3.46, significantly outperforming all benchmarks. Traditional parametric methods struggle severely: LR (38.23% MAPE), LogR (38.60%), LCM (43.59%), and Spline (30.36%) all exhibit substantial bias, likely due to their inability to capture complex interactions between user features and treatment levels. The non-parametric GRF performs poorly (40.43% MAPE). This is because tree-based methods rely on step-function splits and they inherently struggle to extrapolate or interpolate across wide, unobserved gaps in the continuous treatment space when an entire arm is held out. Even PDL (7.18% MAPE), which employs deep learning, achieves 3.2 times worse accuracy than DLPT. This performance gap demonstrates that our doubly robust bias correction procedure provides substantial gains over standard deep learning approaches. The superior extrapolation accuracy validates that our semi-parametric framework effectively learns the underlying dose-response function.

Table 2: ATE Estimation Accuracy Across Held-Out Treatment Levels

| Metric | LR | LogR | LCM | PDL | GRF | Spline | DLPT |
|---|---|---|---|---|---|---|---|
| MAPE (%) | 38.23 | 38.60 | 43.59 | 7.18 | 40.43 | 30.36 | **2.25** |
| MSE | 792.99 | 814.35 | 1028.02 | 32.82 | 1047.68 | 611.50 | **3.46** |

Note: Lower values indicate better performance. MAPE and MSE are averaged across four held-out treatment levels.

## 5.3. Policy Learning Results

Beyond point estimation of treatment effects, we evaluate each method's ability to learn optimal personalized policies within the finite discrete policy class, that is, policies restricted to assigning one of the five experimentally observed treatment levels based on user characteristics.

We partition the user population into 2,287 distinct subgroups based on their covariate profiles. To simulate deployment on new user populations, we randomly split these subgroups into training (50%) and testing (50%) sets. Each method learns a policy using only training subgroups, then we evaluate performance on held-out test subgroups. For each test subgroup, we establish the ground truth optimal policy as the treatment arm that achieved the highest empirical average outcome during the experiment. We then measure regret as the percentage difference between this optimal value and the value achieved by each method's learned policy.

Table 3 presents three complementary performance metrics. First, Mean Percentage Regret (MPR) quantifies the average suboptimality of learned policies, weighted by each subgroup's optimal value. DLPT achieves an MPR of just 0.48%, compared to 6.32% for PDL and 6.41% to 10.05% for other benchmarks. Second, Accuracy measures the percentage of subgroups for which each method correctly identifies the optimal arm. DLPT succeeds for 73.08% of subgroups, whereas all benchmarks achieve only 20% to 25%. Third, Weighted Accuracy accounts for subgroup size, giving more weight to larger user segments. Here DLPT reaches 79.83%, versus 13.50% to 33.79% for benchmarks.

These results demonstrate that DLPT effectively captures treatment effect heterogeneity across diverse user characteristics. The substantial performance gap suggests that competing methods either fail to personalize adequately or suffer from insufficient bias correction when leveraging high-dimensional features. The strong performance of DLPT on held-out subgroups validates its ability to generalize personalized policies to new user populations.

Table 3: Policy Learning Performance under Finite Discrete Policy Class

| Metric | LR | LogR | LCM | PDL | GRF | UNI | Spline | DLPT |
|---|---|---|---|---|---|---|---|---|
| MPR (%) | 7.47 | 10.05 | 9.91 | 6.32 | 6.32 | 6.41 | 6.41 | **0.48** |
| Accuracy (%) | 24.55 | 20.85 | 20.19 | 25.50 | 24.73 | 24.74 | 24.74 | **73.08** |
| Weighted Acc. (%) | 31.61 | 19.64 | 13.50 | 32.70 | 33.78 | 33.79 | 33.79 | **79.83** |

Note: Lower MPR and higher accuracy indicate better performance. Results averaged over 30 random train-test splits of the 2,287 user subgroups.

## 5.4. Continuous Policy Learning Results

Beyond the discrete policy results presented above, we demonstrate that DLPT successfully learns continuous policies that map user features to treatment values over the full range $[0, 1000]$. Using semi-synthetic data with a known ground truth data generation process (polynomial degree $K = 4$), we evaluate policy learning performance across both discrete and continuous policy classes. We hold the training set $\mathcal{S}_{\text{train}}^{\text{FNN}}$ fixed and draw 30 bootstrap resamples from the inference and evaluation sets to assess robustness.

We evaluate performance across four distinct policy classes:

- **Finite Observed**: Policies restricted to the five experimentally observed treatment levels ($\{0, 178, 258, 338, 738\}$).

- **Discretized Continuous**: Policies that select from 101 evenly spaced treatment levels spanning $[0, 1000]$.

- **Linear**: Policies of the form $\pi(\boldsymbol{x}) = \max\{0, \min\{1000, \boldsymbol{w}'\boldsymbol{x}\}\}$, where treatments vary linearly with user features.

- **DNN**: Policies learned via deep neural networks, $\pi(\boldsymbol{x}) = f_{\text{DNN}}(\boldsymbol{x})$, allowing fully flexible nonlinear mapping from features to continuous treatments.

Table 4 reports MPR across all methods for these four policy classes. DLPT significantly outperforms all benchmarks across all policy classes (all pairwise comparisons yield $p < 0.05$). Notably, continuous policy classes (Linear and DNN) generally achieve lower regret than discrete ones across all methods, suggesting that leveraging a richer treatment space enhances policy learning when the estimator can effectively capture heterogeneous treatment effects.

Table 4: MPR Comparison Across Policy Classes and Methods

| Policy Class | DLPT | LR | LogR | LCM | PDL | GRF | Spline | UNI |
|---|---|---|---|---|---|---|---|---|
| Finite Observed | **17.96** | 29.89 | 33.18 | 31.04 | 27.58 | 27.92 | 28.00 | 28.00 |
| Discretized Cont. | **19.69** | 33.04 | 34.75 | 33.44 | 29.28 | 33.61 | 32.37 | 32.80 |
| Linear | **0.01** | 6.01 | 6.37 | 6.44 | 0.66 | 6.20 | 1.75 | 1.75 |
| DNN | **0.99** | 12.65 | 9.20 | 13.33 | 7.42 | 11.05 | 8.46 | 8.46 |

Note: All values are percentages (%). DLPT achieves the lowest MPR in each policy class with $p < 0.05$ in all pairwise comparisons against benchmarks.

The continuous DNN policy learned by DLPT achieves remarkably low regret (0.99% MPR), substantially outperforming the discrete Finite Observed policy (17.96% MPR). This 94.5% reduction in regret demonstrates that DLPT effectively extrapolates from discrete experimental data to learn personalized continuous treatment recommendations. The learned policy $\pi(\boldsymbol{x}) = f_{\text{DNN}}(\boldsymbol{x})$ provides fine-grained, individualized treatment values that go well beyond the five discrete levels observed in the experiment.

Comparing across policy classes reveals important insights. First, the performance gap between DLPT and benchmarks widens substantially in richer policy classes. In the Finite Observed class, DLPT reduces regret by 35% relative to the best benchmark (PDL 27.58%). In the DNN class, this improvement increases to 86.7% (PDL 7.42%). This pattern

indicates that DLPT's advantages of flexible neural network estimation combined with doubly robust debiasing become increasingly valuable as the policy space expands.

Second, while continuous policy classes generally outperform discrete ones, this advantage materializes only when paired with sophisticated estimation methods. Simple parametric approaches (LR, LogR, LCM) perform worse in the DNN class than in discrete classes, likely due to model misspecification amplified over the continuous treatment range. In contrast, DLPT achieves near-optimal performance (0.99% regret) by leveraging its debiasing procedure to provide robustness across the full continuous treatment space, including near boundary values where other methods struggle.

# 6. Conclusion

In this paper, we introduce the DLPT framework, a novel approach for estimating continuous policy values and learning personalized continuous policies with observations collected at discrete treatment levels. We provide theoretical guarantees for DLPT, including the asymptotic unbiasedness of the policy value estimator and a $\sqrt{n}$-regret bound for the learned policies. The practical efficacy of DLPT was rigorously tested through a large-scale field experiment on a leading video-sharing platform, where it demonstrated substantial improvements in policy value estimation and policy regret minimization over conventional methods.

Despite these promising results, we note several limitations that present opportunities for future work. First, our current framework relies on a sigmoid-polynomial parametric assumption to model the underlying dose-response curves. Second, the methodology necessitates the availability of discrete RCT data for the initial learning phase. Third, the learned policy is inherently static and operates within a single-period setting. Regarding potential concerns about user preference manipulation, such as long-run preference drift, we acknowledge this as a scope limitation of the current static framework. Finally, our theoretical analysis relies on the Stable Unit Treatment Value Assumption (SUTVA) for A/B tests. In our real-world deployment, SUTVA violations are minimal, as the experiment involves only a small fraction of the total platform traffic, thereby mitigating substantive interference effects.

# Impact Statement

This paper presents work whose goal is to advance the field of machine learning, specifically in causal inference and offline policy learning. There are many potential societal consequences of our work, none of which we feel must be specifically highlighted here.

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

# Online Appendices

## A. Literature Summary

Table 5: Summary of Related Literature in Personalized Pricing and Policy Learning

| Paper | Data Used | Personalization | Causality | Policy Type | Validation |
|---|---|---|---|---|---|
| *Personalized Pricing* | | | | | |
| (Rossi et al., 1996) | Observational Data | ✓ | NA | Discrete | NA |
| (Chen & Gallego, 2021) | NA | ✓ | NA | Continuous | NA |
| (Qi et al., 2022) | NA | ✓ | ✓ | Continuous | Simulation Data |
| (Smith et al., 2023) | Observational Data | ✓ | NA | Discrete | Cross-Validation |
| (Yoganarasimhan et al., 2023) | Experimental Data | ✓ | ✓ | Discrete | Cross-Validation |
| (Dubé & Misra, 2023) | Experimental Data | ✓ | ✓ | Discrete | Another Experiment Data |
| (Hitsch et al., 2024) | Experimental Data | ✓ | ✓ | Discrete | Another Experiment Data |
| (Aryal et al., 2024) | Observational Data | ✓ | NA | Discrete | NA |
| *Policy Learning* | | | | | |
| (Zhao et al., 2012; 2015) | Experimental Data | ✓ | NA | Discrete | Cross-Validation |
| (Chen et al., 2016) | NA | ✓ | NA | Continuous | Observational Data |
| (Kallus & Zhou, 2018) | NA | ✓ | NA | Continuous | Observational Data |
| (Chernozhukov et al., 2019) | NA | ✓ | NA | Continuous | Simulation Data |
| (Athey & Wager, 2021) | Observational Data | ✓ | NA | Discrete | NA |
| (Zhou et al., 2023) | Observational Data | ✓ | NA | Discrete | NA |

## B. Technical Details

### B.1. Assumptions

**Assumption B.1.** Following Assumption 1 in (Farrell et al., 2020), we require Lipschitz continuity and sufficient curvature, near the truth, for loss function $\ell$ in Stage 1 nuisance parameter estimation.

(a) (LIPSCHITZ CONTINUITY) There exists a positive constant $C_\ell$ such that, for any $\boldsymbol{\theta}(\cdot)$, $\tilde{\boldsymbol{\theta}}(\cdot)$ and $\boldsymbol{x}$,

$$| \ell(y, t, \boldsymbol{\theta}(\boldsymbol{x})) - \ell(y, t, \tilde{\boldsymbol{\theta}}(\boldsymbol{x}))| \leq C_\ell \|\boldsymbol{\theta}(\boldsymbol{x}) - \tilde{\boldsymbol{\theta}}(\boldsymbol{x})\|_2, \qquad (21)$$

(b) (SUFFICIENT CURVATURE) There exist positive constants $c_1$ and $c_2$ such that, for any $\boldsymbol{\theta}(\cdot) \in \mathcal{F}_{DNN}$,

$$c_1 \mathbb{E}[\|\boldsymbol{\theta}(\boldsymbol{X}) - \boldsymbol{\theta}^*(\boldsymbol{X})\|_2^2] \leq \mathbb{E}[\ell(Y, T, \boldsymbol{\theta}(\boldsymbol{X}))] - \mathbb{E}[\ell(Y, T, \boldsymbol{\theta}_K^*(\boldsymbol{x}))] \leq c_2 \mathbb{E}[\|\boldsymbol{\theta}(\boldsymbol{X}) - \boldsymbol{\theta}^*(\boldsymbol{X})\|_2^2]. \qquad (22)$$

**Assumption B.2.** Following Assumption 2 in (Farrell et al., 2020), we make the regularity assumptions on samples and nuisance parameter function $\boldsymbol{\theta}^*(\boldsymbol{x})$:

*(a)* $\boldsymbol{Z}_i = (\boldsymbol{X}_i', T_i, Y_i)'$, $1 \leq i \leq n$, are i.i.d. copies from the population random variables $\boldsymbol{Z} = (\boldsymbol{X}', T, Y)' \in [-1, 1]^{d_{\boldsymbol{X}}} \times \mathcal{T} \times \mathcal{Y}$, where $\mathcal{Y}$ is the bounded support of the outcome $Y$.

*(b)* The parameter function $\boldsymbol{\theta}_K^*(\boldsymbol{x})$ is uniformly bounded. Furthermore, $\theta_k^*(\boldsymbol{x}) \in W^{p,\infty}([-1,1]^{d_{\boldsymbol{X}}})$, $k = 1, 2, \ldots, d_{\boldsymbol{\theta}}$, where for positive integers $p$, define the Sobolev ball $W^{p,\infty}([-1,1]^{d_{\boldsymbol{X}}})$ of functions $h : \mathbb{R}^{d_{\boldsymbol{X}}} \mapsto \mathbb{R}$ with smoothness $p \in \mathbb{N}_+$ as,

$$W^{p,\infty}([-1,1]^{d_{\boldsymbol{X}}}) := \Big\{h : \max_{\boldsymbol{r}, |\boldsymbol{r}| < p} \operatorname*{ess\,sup}_{\boldsymbol{v} \in [-1,1]^{d_{\boldsymbol{X}}}} |D^{\boldsymbol{r}} h(\boldsymbol{v})| \leq 1\Big\},$$

where $\boldsymbol{r} = (r_1, \ldots, r_{d_{\boldsymbol{X}}})$, $|\boldsymbol{r}| = r_1 + \cdots + r_{d_{\boldsymbol{X}}}$ and $D^{\boldsymbol{r}} h$ is the weak derivative.

**Assumption B.3.** The following regularity conditions hold uniformly on the distribution of $\boldsymbol{Z} = (\boldsymbol{X}', T, Y)'$:

(a) $\boldsymbol{\Lambda}(\boldsymbol{x}) = \mathbb{E}[\ell_{\boldsymbol{\theta\theta}}(y, t, \boldsymbol{\theta}(\boldsymbol{x}))|\boldsymbol{X} = \boldsymbol{x}]$ is invertible with bounded inverse. Specifically, $\boldsymbol{\Lambda}(\boldsymbol{x}) = \mathbb{E}[G_{\boldsymbol{\theta}}(\hat{\boldsymbol{\theta}}(\boldsymbol{x}), t)G_{\boldsymbol{\theta}}((\hat{\boldsymbol{\theta}}(\boldsymbol{x}), t)'|\boldsymbol{X} = \boldsymbol{x}]$ if using MSE as loss function in Stage 1 and $\boldsymbol{\Lambda}(\boldsymbol{x}) = \mathbb{E}[G_{\boldsymbol{\theta}}(\hat{\boldsymbol{\theta}}(\boldsymbol{x}), t)\tilde{\boldsymbol{T}}_K'|\boldsymbol{X} = \boldsymbol{x}]$ if using binary cross entropy as loss function in Stage 1.

(b) Value function $H(\boldsymbol{\theta}(\boldsymbol{x}), \pi(\boldsymbol{x}))$ is identified and pathwise differentiable.

**Assumption B.4.** (HÖLDER CONTINUITY OF LATENT FUNCTION) Consider the oracle DGP $f : \mathcal{X} \times \mathcal{T} \to [0, 1]$ such that $f(\boldsymbol{x}, t) = \mathbb{E}[Y|\boldsymbol{X} = \boldsymbol{x}, T = t] = \sigma(\tilde{f}(\boldsymbol{x}, t))$, where $\sigma(z) = \frac{1}{1+e^{-z}}$ is the sigmoid function, and $\tilde{f} : \mathcal{X} \times \mathcal{T} \to \mathbb{R}$ is a latent function. It satisfies that, for all $\boldsymbol{x} \in \mathcal{X}$, the latent function $t \to \tilde{f}(\boldsymbol{x}, t)$ is $s_t$-Hölder continuous with a constant $L > 0$ that does not depend on $\boldsymbol{x}$. Specifically, there exist $L > 0$ and $s_t$ such that for all $\boldsymbol{x} \in \mathcal{X}$ and for all $t, t' \in \mathcal{T}$,

$$|\tilde{f}(\boldsymbol{x}, t) - \tilde{f}(\boldsymbol{x}, t')| \le L|t - t'|^{s_t}, \tag{23}$$

where $s_t$ (with $0 \le s_t \le 1$ typically) measures the smoothness of $\tilde{f}(x, \cdot)$.

## B.2. Proof of Proposition 3.1

We first prove the nonparametric identifiability of nuisance parameter $\boldsymbol{\theta}^*(\boldsymbol{x})$ in Proposition 3.1(a). Following DGP defined in Equation 5, let $G(\boldsymbol{\theta}^*(\boldsymbol{x}), t) = G(\tilde{\boldsymbol{\theta}}(\boldsymbol{x}), t)$. To show $\boldsymbol{\theta}^*(\boldsymbol{x})$ can be nonparametrically identified, we show that $\boldsymbol{\theta}^*(\boldsymbol{x}) = \tilde{\boldsymbol{\theta}}(\boldsymbol{x})$. Since $G(\boldsymbol{\theta}^*(\boldsymbol{x}), t) = G(\tilde{\boldsymbol{\theta}}(\boldsymbol{x}), t)$, and the sigmoid function is reversible, we have that

$$\theta_0^*(\boldsymbol{x}) + \theta_1^*(\boldsymbol{x})t + \theta_2^*(\boldsymbol{x})t^2 + \theta_3^*(\boldsymbol{x})t^3 = \tilde{\theta}_0(\boldsymbol{x}) + \tilde{\theta}_1(\boldsymbol{x})t + \tilde{\theta}_2(\boldsymbol{x})t^2 + \tilde{\theta}_3(\boldsymbol{x})t^3$$
$$(\boldsymbol{\theta}^*(\boldsymbol{x}) - \tilde{\boldsymbol{\theta}}(\boldsymbol{x}))(1, t, t^2, t^3)' = 0$$

Define $\tilde{\boldsymbol{T}} = (1, t, t^2, t^3)$, we therefore have

$$(\boldsymbol{\theta}^*(\boldsymbol{x}) - \tilde{\boldsymbol{\theta}}(\boldsymbol{x}))\tilde{\boldsymbol{T}}' = 0$$
$$\mathbb{E}[(\boldsymbol{\theta}^*(\boldsymbol{X}) - \tilde{\boldsymbol{\theta}}(\boldsymbol{X})\tilde{\boldsymbol{T}})^2|\boldsymbol{X}] = (\boldsymbol{\theta}^*(\boldsymbol{X}) - \tilde{\boldsymbol{\theta}}(\boldsymbol{X}))\mathbb{E}[\tilde{\boldsymbol{T}}\tilde{\boldsymbol{T}}'|\boldsymbol{X}](\boldsymbol{\theta}^*(\boldsymbol{X}) - \tilde{\boldsymbol{\theta}}(\boldsymbol{X}))' = 0$$

Since $\mathbb{E}[\tilde{\boldsymbol{T}}\tilde{\boldsymbol{T}}'|\boldsymbol{X}]$ is positive definite uniformally with respect to $\boldsymbol{X}$, we have that $\boldsymbol{\theta}^*(\boldsymbol{X}) - \tilde{\boldsymbol{\theta}}(\boldsymbol{X}) = 0$, i.e., $\boldsymbol{\theta}^*(\boldsymbol{x})$ can be nonparametrically identified.

We proceed to prove the convergence rate of nuisance parameter estimators in Proposition 3.1(b). We use the following result in (Farrell et al., 2020).

**Lemma B.5** (Theorem 1 in (Farrell et al., 2020)). *Suppose Assumptions B.1 and B.2 hold. With the structured DNN of width $H = O(n^{\frac{d_{\boldsymbol{X}}}{2(p+d_{\boldsymbol{X}})}} \log^2 n)$ and depth $L = O(\log n)$, there exists a constant $C$ such that*

$$\|\hat{\boldsymbol{\theta}}_k - \boldsymbol{\theta}_k^*\|_{L_2(\boldsymbol{X})}^2 \lesssim n^{-\frac{p}{p+d_{\boldsymbol{X}}}} \log^8 n + \frac{\log \log n}{n}$$

*and*

$$\mathbb{E}_n[(\hat{\boldsymbol{\theta}}_k - \boldsymbol{\theta}_k^*)^2] \lesssim n^{-\frac{p}{p+d_{\boldsymbol{X}}}} \log^8 n + \frac{\log \log n}{n}$$

*for $n$ large enough with probability at least $1 - \exp(n^{-\frac{d_{\boldsymbol{X}}}{p+d_{\boldsymbol{X}}}} \log^8 n)$, for $k = 1, \dots, d_{\boldsymbol{\theta}}$.*

To achieve the convergence rate in Proposition 3.1(b), we need to verify assumptions made in Lemma B.5.

1. The nonparametric identifiability of $\boldsymbol{\theta}^*(\boldsymbol{x})$ can be satisfied as long as $\mathbb{E}[\tilde{\boldsymbol{T}}\tilde{\boldsymbol{T}}'|\boldsymbol{X}]$ is positive definite uniformly with respect to $\boldsymbol{X}$, which is satisfied in our RCT setting when the number of discrete levels of treatment $m >= 4$.

2. The Lipschitz continuity of loss function $\ell$ is satisfied under either MSE loss function or binary cross-entropy loss function since $G(\cdot)$ is a sigmoid function with bounded random variable $\boldsymbol{Z}$.

3. The curvature condition of the loss function near the truth can be verified through the following derivations. When using MSE as loss function:

$$\mathbb{E}[\ell(Y, T, \boldsymbol{\theta}(\boldsymbol{X}))] - \mathbb{E}[\ell(Y, T, \boldsymbol{\theta}_K^*(\boldsymbol{X}))] = \mathbb{E}\left[(G(\boldsymbol{\theta}(\boldsymbol{X}), T) - G(\boldsymbol{\theta}^*(\boldsymbol{X}), T))^2\right]$$

Since $G(\cdot)$ is a continuous and differentiable with respect to $\boldsymbol{\theta}(\boldsymbol{x})$, we can find $\boldsymbol{\theta}_0(\boldsymbol{x})$ which has $\theta_{0k}(\boldsymbol{x}) \in (\theta_k(\boldsymbol{x}), \theta_k^*(\boldsymbol{x}))$ for each $k$ satisfies that $G_{\boldsymbol{\theta}}(\boldsymbol{\theta}_0(\boldsymbol{x}), t) = \frac{G(\boldsymbol{\theta}(\boldsymbol{x}), t) - G(\boldsymbol{\theta}^*(\boldsymbol{x}), t)}{\boldsymbol{\theta}(\boldsymbol{x}) - \boldsymbol{\theta}^*(\boldsymbol{x})}$. Therefore, we have

$$\mathbb{E}[\ell(Y, T, \boldsymbol{\theta}(\boldsymbol{X}))] - \mathbb{E}[\ell(Y, T, \boldsymbol{\theta}_K^*(\boldsymbol{x}))] = \mathbb{E}\left[(\boldsymbol{\theta}(\boldsymbol{X}) - \boldsymbol{\theta}^*(\boldsymbol{X}))G_{\boldsymbol{\theta}}(\boldsymbol{\theta}_0(X), T)G_{\boldsymbol{\theta}}(\boldsymbol{\theta}_0(X), T)'(\boldsymbol{\theta}(\boldsymbol{X}) - \boldsymbol{\theta}^*(\boldsymbol{X}))'\right] \tag{24}$$

$$= \mathbb{E}[((\boldsymbol{\theta}(\boldsymbol{X}) - \boldsymbol{\theta}^*(\boldsymbol{X}))\tilde{\boldsymbol{T}})^2 G(\boldsymbol{\theta}_0(X), T)^2(1 - G(\boldsymbol{\theta}_0(X), T))^2] \tag{25}$$

From Equation (24), we verify that the second inequality in the curvature condition is satisfied since $G(\cdot)$ is bounded away from 0 and 1 by the boundedness of the covariate space and treatment space. Deriving from Equation (25), we further have that

$$
\begin{aligned}
\mathbb{E}[\ell(Y, T, \boldsymbol{\theta}(\boldsymbol{X}))] - \mathbb{E}[\ell(Y, T, \boldsymbol{\theta}_K^*(\boldsymbol{x}))] &\geq c\mathbb{E}[((\boldsymbol{\theta}(\boldsymbol{X}) - \boldsymbol{\theta}^*(\boldsymbol{X}))\tilde{\boldsymbol{T}})^2] \\
&\geq d\mathbb{E}[(\boldsymbol{\theta}^*(\boldsymbol{X}) - \tilde{\boldsymbol{\theta}}(\boldsymbol{X}))\mathbb{E}[\tilde{\boldsymbol{T}}\tilde{\boldsymbol{T}}'|\boldsymbol{X}](\boldsymbol{\theta}^*(\boldsymbol{X}) - \tilde{\boldsymbol{\theta}}(\boldsymbol{X}))'] \\
&\geq e\mathbb{E}[(\boldsymbol{\theta}^*(\boldsymbol{X}) - \tilde{\boldsymbol{\theta}}(\boldsymbol{X})(\boldsymbol{\theta}^*(\boldsymbol{X}) - \tilde{\boldsymbol{\theta}}(\boldsymbol{X}))']
\end{aligned}
$$

which proves that the first inequality in the curvature condition is satisfied.

To this end, we conclude the proof for Proposition 3.1.

## B.3. Proof of Proposition 3.3

We derive the influence function in Proposition 3.3 following Theorem 2 in (Farrell et al., 2020).

**Lemma B.6** (Theorem 2 in (Farrell et al., 2020))**.** *The following conditions hold uniformly in the given conditioning elements. (i) DGP (5) holds and identifies $\boldsymbol{\theta}^*(\cdot)$. (ii) $\mathbb{E}[\ell_{\boldsymbol{\theta}}(y, t, \boldsymbol{\theta}^*(\boldsymbol{x}))|\boldsymbol{X} = \boldsymbol{x}, T = t] = 0$. (iii) $\boldsymbol{\Lambda}(x) := \mathbb{E}[\ell_{\boldsymbol{\theta}\boldsymbol{\theta}}(Y, T, \boldsymbol{\theta}(\boldsymbol{x}))|\boldsymbol{X} = \boldsymbol{x}]$ is invertible with uniformly bounded inverse. (iv) Parameter $V_K(\pi)$ is identified, pathwise differentiable, and $H$ and $\ell$ are thrice continuously differentiable in $\boldsymbol{\theta}$. (v) $H(\boldsymbol{\theta}^*(\boldsymbol{X}), \pi(\boldsymbol{x}))$ and $\ell_{\boldsymbol{\theta}}(Y, T, \boldsymbol{\theta}^*(\boldsymbol{X}))$ possess $q > 4$ finite absolute moments and positive variance. Then for the policy value $Q(\pi)$, the Neyman orthogonal score is $\psi(\boldsymbol{z}, \pi; \boldsymbol{\theta}, \boldsymbol{\Lambda}) - V_K(\pi)$, where*

$$
\psi(\boldsymbol{z}, \pi; \hat{\boldsymbol{\theta}}, \boldsymbol{\Lambda}) = H(\hat{\boldsymbol{\theta}}(\boldsymbol{x}), \pi(\boldsymbol{x})) - H_{\boldsymbol{\theta}}(\hat{\boldsymbol{\theta}}(\boldsymbol{x}), \pi(\boldsymbol{x}))\boldsymbol{\Lambda}(\boldsymbol{x})^{-1}\ell_{\boldsymbol{\theta}}(y, t, \hat{\boldsymbol{\theta}}(\boldsymbol{x})), \tag{26}
$$

*where $\ell_{\boldsymbol{\theta}}, H_{\boldsymbol{\theta}}$ are $d_{\boldsymbol{\theta}}$-dimensional vectors of first order derivatives, and $\ell_{\boldsymbol{\theta}\boldsymbol{\theta}}$ is the $d_{\boldsymbol{\theta}} \times d_{\boldsymbol{\theta}}$ Hessian matrix of $\ell$, with $\{k_1, k_2\}$ element defined by $\partial^2 \ell/\partial \theta_{k_1} \partial \theta_{k_2}$.*

To obtain the influence function for DLPT policy value estimator $\hat{V}_K^{\mathsf{DLPT}}(\pi)$, we need to verify assumptions made in Lemma B.6.

1. The nonparametric identifiability of $\boldsymbol{\theta}^*(\boldsymbol{x})$ can be satisfied as long as $\mathbb{E}[\tilde{\boldsymbol{T}}\tilde{\boldsymbol{T}}'|\boldsymbol{X}]$ is positive definite uniformly with respect to $\boldsymbol{X}$, which is satisfied in our RCT setting when the number of discrete levels of treatment $m >= 4$.

2. $\ell_{\boldsymbol{\theta}}(y, t, \boldsymbol{\theta}^*(\boldsymbol{x})) = G_{\boldsymbol{\theta}}(\boldsymbol{\theta}^*(\boldsymbol{x}), t)(G(\boldsymbol{\theta}^*(\boldsymbol{x}), t) - y)$ if using MSE as loss function in Stage 1, and $\ell_{\boldsymbol{\theta}}(y, t, \boldsymbol{\theta}^*(\boldsymbol{x})) = (G(\boldsymbol{\theta}^*(\boldsymbol{x}), t) - y)\tilde{T}$ if using binary cross entropy as loss function in Stage 1. (ii) holds under both loss functions with DGP assumption in Equation (5).

3. $\boldsymbol{\Lambda}(\boldsymbol{x}) = \mathbb{E}[G_{\boldsymbol{\theta}}((\hat{\boldsymbol{\theta}}(\boldsymbol{x}), t)G_{\boldsymbol{\theta}}((\hat{\boldsymbol{\theta}}(\boldsymbol{x}), t)'|\boldsymbol{X} = \boldsymbol{x}]$ if using MSE as loss function in Stage 1 and $\boldsymbol{\Lambda}(\boldsymbol{x}) = \mathbb{E}[G_{\boldsymbol{\theta}}((\hat{\boldsymbol{\theta}}(\boldsymbol{x}), t)\tilde{T}']$ if using binary cross entropy as loss function in Stage 1. (iii) holds under both loss functions as $G(\cdot)$ is bounded away from 0 and 1 with the boundedness of covariate and treatment spaces, as well as $G_{\boldsymbol{\theta}} = G(1 - G)\tilde{T}$.

4. (iv) holds as both $H$ and $\ell$ enjoy sufficient smoothness.

5. (v) holds as $G(\cdot)$ enjoys sufficient smoothness with bounded value of nuisance parameter.

To this end, we conclude our proof of Proposition 3.3.

## B.4. Proof of Proposition 3.5

To show the universal orthogonality, we conduct the following derivation:

$$
\begin{aligned}
&\mathbb{E}[\nabla_{\boldsymbol{\theta}}\psi(\boldsymbol{z}, \pi; \boldsymbol{\theta}^*, \boldsymbol{\Lambda})|\boldsymbol{X} = \boldsymbol{x}] \\
&= \mathbb{E}\Big[H_{\boldsymbol{\theta}}(\boldsymbol{\theta}(\boldsymbol{x}), \pi(\boldsymbol{x})) - H_{\boldsymbol{\theta}}(\boldsymbol{\theta}(\boldsymbol{x}), \pi(\boldsymbol{x}))\boldsymbol{\Lambda}(\boldsymbol{x})^{-1}\ell_{\boldsymbol{\theta}\boldsymbol{\theta}}(y, t, \boldsymbol{\theta}(\boldsymbol{x})) - H_{\boldsymbol{\theta}\boldsymbol{\theta}}(\boldsymbol{\theta}(\boldsymbol{x}), \pi(\boldsymbol{x}))\boldsymbol{\Lambda}(\boldsymbol{x})^{-1}\ell_{\boldsymbol{\theta}}(y, t, \boldsymbol{\theta}(\boldsymbol{x}))\Big|\boldsymbol{X} = \boldsymbol{x}\Big] \\
&\stackrel{(i)}{=} \mathbb{E}\Big[-H_{\boldsymbol{\theta}\boldsymbol{\theta}}(\boldsymbol{\theta}(\boldsymbol{x}), \pi(\boldsymbol{x}))\boldsymbol{\Lambda}(\boldsymbol{x})^{-1}\ell_{\boldsymbol{\theta}}(y, t, \boldsymbol{\theta}(\boldsymbol{x}))\Big|\boldsymbol{X} = \boldsymbol{x}\Big] \\
&= -H_{\boldsymbol{\theta}\boldsymbol{\theta}}(\boldsymbol{\theta}(\boldsymbol{x}), \pi(\boldsymbol{x}))\boldsymbol{\Lambda}(\boldsymbol{x})^{-1}\mathbb{E}\Big[\ell_{\boldsymbol{\theta}}(y, t, \boldsymbol{\theta}(\boldsymbol{x}))\Big|\boldsymbol{X} = \boldsymbol{x}\Big] \stackrel{(ii)}{=} 0
\end{aligned}
$$

where (i) is by definition that $\boldsymbol{\Lambda}(x) := \mathbb{E}[\ell_{\boldsymbol{\theta}\boldsymbol{\theta}}(Y, T, \boldsymbol{\theta}(\boldsymbol{x}))|\boldsymbol{X} = \boldsymbol{x}]$, and (ii) is by the first order optimality of $\boldsymbol{\theta}(\cdot)$.

## B.5. Proof of Theorem 3.7

We first prove a more involved version of the regret bound, which will imply Theorem 3.7 as shown in Appendix B.5.2. To establish this new regret bound, below we introduce a few more technical terms to characterize the policy class complexity:

**Definition B.7** (Covering Number and Entropy Integral). *For a function $f : \mathcal{X} \to R$, denote its $L_2$-norm as $\|f\|_2 = \sqrt{\mathbb{E}[f(x)^2]}$ and its empirical $L_2$-norm as $\|f\|_{2,n} = \sqrt{\mathbb{E}_n[f(x)^2]}$. For a real-valued function class $\mathcal{F}$, define the following properties to measure its complexity:*

- *Empirical covering number $\mathcal{N}_2(\epsilon, \mathcal{F}, x_{1:n})$: the size of the smallest function class $\mathcal{F}_\epsilon$, where $\mathcal{F}_\epsilon$ is the $\epsilon$-cover of $\mathcal{F}$ such that for any $f \in \mathcal{F}$, there exists an $f_\epsilon \in \mathcal{F}_\epsilon$ that satisfies $\|f - f_\epsilon\|_{2,n} \leq \epsilon$.*

- *Covering number $\mathcal{N}_2(\epsilon, \mathcal{F}, n)$: the maximal empirical covering number over all possible $n$ samples.*

- *Entropy integral $\kappa(r, \mathcal{F}) = \inf_{\alpha \geq 0} \left\{ 4\alpha + 10 \int_\alpha^r \sqrt{\frac{\log \mathcal{N}_2(\epsilon, \mathcal{F}, n)}{n}} d\epsilon \right\}$.*

We now introduce the following regret bound.

**Proposition B.8.** *Let $S_1$ and $S_2$ be randomly and evenly splits of $n$ i.i.d. samples. Let $\hat{\boldsymbol{\theta}}$ be the estimated nuisance on sample $S_1$ following procedure in Stage 1. Suppose that $\|\hat{\boldsymbol{\theta}} - \boldsymbol{\theta}^*\|_{L_2(\boldsymbol{X})} = o(n^{-1/4})$, which is guaranteed by Proposition 3.1. Let $\hat{\pi}$ be the learned policy that solves the ERM problem: $\hat{\pi} := \inf_{\pi \in \Pi} \left\{ -\mathbb{E}_{S_2}[\psi(z, \pi; \hat{\boldsymbol{\theta}}, \Lambda)] \right\}$, where $\mathbb{E}_{S_2}[\cdot]$ denotes the empirical average over sample $S_2$. Define the function class: $\psi \circ \Pi = \{\psi(\cdot, \pi; \hat{\boldsymbol{\theta}}, \Lambda) : \pi \in \Pi\}$. Let $r = \sup_{\pi \in \Pi} \|\psi(\cdot, \pi; \hat{\boldsymbol{\theta}}, \Lambda) - \psi(\cdot, \pi^*; \hat{\boldsymbol{\theta}}, \Lambda)\|_{L_2(\boldsymbol{X})}$. Assume Assumptions B.1, B.2, and B.3 in Appendix B.1 hold, then with probability $1 - \delta$,*

$$R_K(\hat{\pi}) = O\left( \kappa(r, \psi \circ \Pi) + \frac{\log \mathcal{N}_2(r, \psi \circ \Pi, n)}{n} + r\sqrt{\frac{\log(1/\delta)}{n}} + \frac{\log(1/\delta)}{n} \right). \tag{27}$$

In the following, we shall first prove Proposition B.8 in Appendix B.5.1 and then prove Theorem 3.7 in Appendix B.5.2.

### B.5.1. PROVING PROPOSITION B.8.

The following two lemmas form a building block in our proof.

**Lemma B.9** ((Foster & Syrgkanis, 2019), Theorem 2). *Let $S = \{(\boldsymbol{z}_i)_{i=1}^n\}$ be $n$ i.i.d. observations sampled from space $\mathcal{Z}$ under distribution $\mathcal{D}$. Denote $\mathcal{F}$ as the function class of the decision variable and $\mathcal{G}$ as the function class of a nuisance model. Let $\ell : \mathcal{Z} \times \mathcal{F} \times \mathcal{G} \to R$ be a loss function. Let $S_1$ and $S_2$ be two evenly and randomly splits of sample $S$. Let $\hat{g} \in \mathcal{G}$ be the estimated nuisance on $S_1$ and $g^* \in \mathcal{G}$ be the true nuisance parameter. Let $\hat{f} = \inf_{f \in \mathcal{F}} \mathbb{E}_{S_2}[\ell(\boldsymbol{z}, f, \hat{g})]$ be the solution to the constrained ERM, where $\mathbb{E}_{S_2}[\cdot]$ denotes the sample average on sample $S_2$.*

*Let $L_\mathcal{D}(f, g)$ denote the population loss with respect to $(f, g)$ when marginalized over the sample space $\mathcal{Z}$ under distribution $\mathcal{D}$. Suppose $L_\mathcal{D}(f, g)$ satisfies universal orthogonality condition with respect to the nuisance parameter $g$. Suppose that $L_\mathcal{D}(f, g)$ are thrice-continuously differentiable and that $L_\mathcal{D}(f, g)$ has bounded curvature with respect to $g$. Then the constrained ERM $\hat{f}$ has the following excess risk bound:*

$$L_\mathcal{D}(\hat{f}, g^*) - L_\mathcal{D}(f^*, g^*) = O_p\left( \{L_\mathcal{D}(\hat{f}, \hat{g}) - L_\mathcal{D}(f^*, \hat{g})\} + \|\hat{g} - g^*\|_{L_2(\boldsymbol{Z})}^2 \right), \tag{28}$$

*where $f^* =_{f \in \mathcal{F}} L_\mathcal{D}(f, g^*)$.*

Lemma B.9 shows that, with universal orthogonal loss functions, the regret bound can be decomposed into (i) the regret bound evaluated at the estimated nuisance and (ii) the nuisance estimation error. The following result quantifies the first component.

**Lemma B.10** ((Foster & Syrgkanis, 2019), Theorem 4; (Chernozhukov et al., 2019), Theorem 4). *Follow the notations in Lemma B.9. Define the function class $\ell \circ \mathcal{F} := \{\ell(\theta(\cdot), \hat{g}_n(\cdot), \cdot) : f \in \mathcal{F}_n\}$. Let $r = \sup_{f \in \mathcal{F}} \|\ell(\cdot, f, \hat{g}) - \ell(\cdot, f^*, \hat{g})\|_{L_2(\boldsymbol{Z})}$. Then with probability $1 - \delta$,*

$$L_\mathcal{D}(\hat{f}, \hat{g}) - L_\mathcal{D}(f^*, \hat{g}) = O\left( \kappa(r, \ell \circ \mathcal{F}) + \frac{\log \mathcal{N}_2(r, \ell \circ \mathcal{F}, n)}{n} + r\sqrt{\frac{\log(1/\delta)}{n}} + \frac{\log(1/\delta)}{n} \right). \tag{29}$$

Recall that in our settings, $\psi()$ is universally orthogonal with respect to the nuisance $\boldsymbol{\theta}$, by Proposition 3.5. We also assume that the nuisance component admits the following estimation rate: $\|\hat{\boldsymbol{\theta}} - \boldsymbol{\theta}^*\|_{L_2(\boldsymbol{X})} = o(n^{-1/4})$, which is also characterized in Proposition 3.1. Moreover, Assumption B.1 gives us the bounded curvature. Combining Lemma B.9 and Lemma B.10, we have that with probability $1 - \delta$,

$$
\begin{aligned}
R_K(\hat{\pi}) &= O\Bigg( \kappa(r, \psi \circ \Pi) + \frac{\log \mathcal{N}_2(r, \psi \circ \Pi, n)}{n} + r\sqrt{\frac{\log(1/\delta)}{n}} + \frac{\log(1/\delta)}{n} + \|\hat{\boldsymbol{\theta}} - \boldsymbol{\theta}^*\|_{L_2(\boldsymbol{X})}^2 \Bigg) \\
&= O\Bigg( \kappa(r, \psi \circ \Pi) + \frac{\log \mathcal{N}_2(r, \psi \circ \Pi, n)}{n} + r\sqrt{\frac{\log(1/\delta)}{n}} + \frac{\log(1/\delta)}{n} \Bigg) + o(n^{-1/2}) \\
&= O\Bigg( \kappa(r, \psi \circ \Pi) + \frac{\log \mathcal{N}_2(r, \psi \circ \Pi, n)}{n} + r\sqrt{\frac{\log(1/\delta)}{n}} + \frac{\log(1/\delta)}{n} \Bigg),
\end{aligned}
$$

completing the proof.

### B.5.2. PROVING THEOREM 3.7.

Now we employ Proposition B.8 to show Theorem 3.7. Consider $\psi \circ \Pi$ as a function class with VC subgraph dimension as $d$. Then we have $\mathcal{N}_2(\epsilon, \psi \circ \Pi, n) = O\big( \exp(d(1 + \log(1/\epsilon))) \big)$ (Van der Vaart, 2000). We therefore have

$$
\begin{aligned}
\kappa(r, \psi \circ \Pi) &\leq 10 \int_0^r \sqrt{\frac{d(1 + \log(1/\epsilon))}{n}} \mathrm{d}\epsilon \\
&= 10\epsilon \sqrt{\frac{d(1 + \log(1/\epsilon))}{n}} \Big|_0^r + 10 \int_0^r \sqrt{\frac{d}{n}} \frac{1}{2\sqrt{1 + \log(1/\epsilon)}} \mathrm{d}\epsilon \\
&= r\sqrt{d/n}(1 + \sqrt{1 + \log(1/r)}).
\end{aligned}
$$

By Proposition B.8, we have the regret bound as

$$
\begin{aligned}
R_K(\hat{\pi}) &= O\Bigg( \kappa(r, \psi \circ \Pi) + \frac{\mathcal{N}_2(r, \psi \circ \Pi, n)}{n} + r\sqrt{\frac{\log(1/\delta)}{n}} + \frac{\log(1/\delta)}{n} \Bigg) \\
&= O\Bigg( r\sqrt{d/n}(1 + \sqrt{1 + \log(1/r)}) + \frac{d(1 + \log(1/r))}{n} + r\sqrt{\frac{\log(1/\delta)}{n}} + \frac{\log(1/\delta)}{n} \Bigg) \\
&= O\Bigg( r(1 + \sqrt{1 + \log(1/r)})\sqrt{\frac{d}{n}} + r\sqrt{\frac{\log(1/\delta)}{n}} \Bigg) \\
&= O\Bigg( r(1 + \sqrt{\log(1/r)})\sqrt{\frac{d}{n}} + r\sqrt{\frac{\log(1/\delta)}{n}} \Bigg),
\end{aligned}
$$

completing our proof.

## C. Empirical Setting and Data Details

### C.1. Platform Context and Incentive Mechanism

We partnered with "Platform O," a leading global short-video platform where users act as both content creators and viewers. While prominent creators monetize content through brand deals, smaller creators—especially newcomers—often lack resources. To incentivize these users, the platform uses a proprietary digital token called "Points," which users can earn through engagement-based initiatives. These Points can be redeemed for cash-equivalent rewards (e.g., shopping coupons). Users access these incentives via a "Task Center" in their mobile wallet.

In our experimental intervention, the platform utilized the "Points Earning Task" feature. Users in the treatment groups saw a specific task titled "Post a Video," which offered a financial incentive (paid in Points) for uploading content. The platform's objective was to identify the optimal cost-effective incentive level to encourage low-activity users to transition into regular content production.

## C.2. Data and Randomization

**Sample Construction.** The experiment focused on creators classified as "low-activity" (users who produced videos on fewer than three days in the four weeks preceding the experiment). This segment accounted for $12.3\%$ of all producers on the platform. Due to data confidentiality, we utilize a subsample of $7,349,648$ users. The specific sample sizes for each group are:

- **Control** ($T = 0$): 1,478,472 users.

- **Treatment** ($T = 178$): 1,478,666 users.

- **Treatment** ($T = 358$): 1,479,277 users.

- **Treatment** ($T = 538$): 1,478,819 users.

- **Treatment** ($T = 718$): 1,479,414 users.

**Outcome Definition.** The outcome $Y_i$ is defined as a binary variable ($Y_i = 1$ if the user posted at least one video, $Y_i = 0$ otherwise). We adopt this definition based on platform feedback: (1) only $9.53\%$ of low-activity users post multiple videos in a week, and (2) the primary business goal is establishing the *habit* of creation, for which a single upload is the key metric.

**User Covariate Data Description.** Table 6 presents all the user covariate data used in our empirical analysis.

**Randomization Check.** To ensure the validity of the randomization process, we compared user demographics, content consumption, and content creation behavior during the 10 days prior to the experiment across all groups. As shown in Table 7, the five groups exhibited similar distributions in key demographic variables (e.g., gender, city tier) and pre-experiment behaviors. All pairwise $t$-tests between control and treatment groups yielded $p$-values $> 0.05$, confirming successful randomization.

# D. Model-Free Empirical Evidence

## D.1. Average Treatment Effects (ATE)

We estimated the ATE for each reward level using the following OLS specification:

$$Y_i = \alpha_0 + \alpha_1 D_i + \epsilon_i, \tag{30}$$

where $Y_i$ is the binary outcome and $D_i$ is the treatment indicator. As shown in Table 1, we observe a clear monotonic increase in effect size as the reward increases. The maximum pairwise $p$-value between adjacent levels is $< 1.189 \times 10^{-7}$, confirming that higher incentives consistently lead to greater engagement, though with diminishing marginal returns.

## D.2. Heterogeneous Treatment Effects (HTE)

We investigated heterogeneity using two moderators: historical production ($LowP_i$) and historical consumption ($LowC_i$). We employed the following interaction specifications:

$$Y_i = \beta_0 + \beta_1 D_i + \beta_2 D_i \times LowP_i + \beta_3 LowP_i + \epsilon_i \tag{31}$$

$$Y_i = \gamma_0 + \gamma_1 D_i + \gamma_2 D_i \times LowC_i + \gamma_3 LowC_i + \epsilon_i \tag{32}$$

Results in Table 8 show that $LowP$ users (low historical production) have negative interaction terms (all $p < 0.0001$), indicating lower responsiveness to incentives. Conversely, $LowC$ users (low historical consumption) have positive interaction terms, indicating higher responsiveness.

# E. Implementation Details

## E.1. DLPT Implementation

We follow the three-stage procedure outlined in Section 3.

Table 6: User Covariates Used in the Empirical Analysis

| | Variable | Description |
|---|---|---|
| Disc-rete Var. | Gender | Gender of user: male, female, or unknown |
| | User Activeness Degree | Activeness of user: high-, mid-, low-active, or new user |
| | User Life Time Stage | Life cycle phase of user on the platform: new, mature, or recession |
| | Operating System | OS of user's device: Android, iPhone, IPAD, or other |
| | Installation of video-sharing platform A | Installed or not |
| | Installation of longer video-sharing platform B | Installed or not |
| | Installation of live-streaming platform C | Installed or not |
| | Installation of game live-streaming platform D | Installed or not |
| | Installation of game live-streaming platform E | Installed or not |
| | Frequent Residence Area | Region in which the user is frequently on the platform: South, North, or unknown |
| | Frequent Residence City Level | Level of the city in which the user is frequently on the platform: large city, big city, medium city, small city, or unknown |
| | Number of Mutual Followers | Interval of the user's number of mutual followers (friends): $<5$, 5 - 30, 30 - 60, 60-120, 120-250 $>250$ |
| Conti-nuous Var. | Age | Age of the user |
| | Number of Followed Users | Number of users followed by the user before the treatment |
| | Number of Fans | Number of users following the user before the treatment |
| | Model Price | The price of the device model used by the user |
| | *In past 30 days on simplified app version* | |
| | Average App Usage Duration | User's average usage duration on platform per day |
| | Average Video Watching Time | User's average time on watching videos on platform per day |
| | Average Profile Page Staying Time | User's average time staying on his own profile page per day |
| | Number of videos being completely watched | Total number of fully viewed videos by the user |
| | Number of videos being validly watched | Total number of videos thoroughly and validly viewed by the user |
| | Number of videos uploading | Total number of videos posted by the user |
| | Number of app launch times uploading | Total number of times of user launching the app |
| | *In past 10 days on simplified app version* | |
| | Average App Usage Duration | User's average usage duration on platform per day |
| | Average Video Watching Time | User's average time on watching videos on platform per day |
| | Average Profile Page Staying Time | User's average time staying on his own profile page per day |
| | Number of videos being completely watched | Total number of fully viewed videos by the user |
| | Number of videos being validly watched | Total number of videos thoroughly and validly viewed by the user |
| | Number of videos uploading | Total number of videos posted by the user |
| | Number of app launch times uploading | Total number of times of user launching the app |
| | *In past 3 days on simplified app version* | |
| | Average App Usage Duration | User's average usage duration on platform per day |
| | Average Video Watching Time | User's average time on watching videos on platform per day |
| | Average Profile Page Staying Time | User's average time staying on his own profile page per day |
| | Number of videos being completely watched | Total number of fully viewed videos by the user |
| | Number of videos being validly watched | Total number of videos thoroughly and validly viewed by the user |
| | Number of videos uploading | Total number of videos posted by the user |
| | Number of app launch times uploading | Total number of times of user launching the app |
| | *In past 30 days on original app version* | |
| | Average App Usage Duration | User's average usage duration on platform per day |
| | Average Video Watching Time | User's average time on watching videos on platform per day |
| | Average Profile Page Staying Time | User's average time staying on his own profile page per day |
| | Number of videos being completely watched | Total number of fully viewed videos by the user |
| | Number of videos being validly watched | Total number of videos thoroughly and validly viewed by the user |
| | Number of videos uploading | Total number of videos posted by the user |
| | Number of app launch times uploading | Total number of times of user launching the app |
| | *In past 10 days on original app version* | |
| | Average App Usage Duration | User's average usage duration on platform per day |
| | Average Video Watching Time | User's average time on watching videos on platform per day |
| | Average Profile Page Staying Time | User's average time staying on his own profile page per day |
| | Number of videos being completely watched | Total number of fully viewed videos by the user |
| | Number of videos being validly watched | Total number of videos thoroughly and validly viewed by the user |
| | Number of videos uploading | Total number of videos posted by the user |
| | Number of app launch times uploading | Total number of times of user launching the app |
| | *In past 3 days on original app version* | |
| | Average App Usage Duration | User's average usage duration on platform per day |
| | Average Video Watching Time | User's average time on watching videos on platform per day |
| | Average Profile Page Staying Time | User's average time staying on his own profile page per day |
| | Number of videos being completely watched | Total number of fully viewed videos by the user |
| | Number of videos being validly watched | Total number of videos thoroughly and validly viewed by the user |
| | Number of videos uploading | Total number of videos posted by the user |
| | Number of app launch times uploading | Total number of times of user launching the app |

**Stage 1: Nuisance Parameter Estimation.** We assume the data generation process (DGP) follows the semi-parametric

Table 7: Randomization Check

|  |  | 0 | 178 | 358 | 538 | 718 |
|---|---|---|---|---|---|---|
| *User Demographics* | Proportion of Female Users | 42.38% | 42.39% (0.95) | 42.48% (0.20) | 42.31% (0.52) | 42.41% (0.52) |
|  | Proportion of High-Active Users | 29.36% | 29.40% (0.49) | 29.38% (0.76) | 29.38% (0.75) | 29.42% (0.23) |
| *User Behaviors (Pre-Exp)* | App Usage Duration | 0.0010 | -0.0000 (0.34) | 0.0005 (0.67) | 0.0011 (0.93) | -0.0027 (0.95) |
|  | Video Upload | -0.0014 | 0.0003 (0.15) | 0.0005 (0.11) | -0.0007 (0.52) | 0.0002 (0.11) |

Note: p-values of t-tests between the control group $t = 0$ and other levels of points are reported in parentheses. Pre-experiment behavior metrics are rescaled to protect sensitive data.

Table 8: Heterogeneous Treatment Effects of Point Rewards

| | Dependent Variable: Video Uploading ($Y$) | | | | | | | |
|---|---|---|---|---|---|---|---|---|
| | 178 | | 358 | | 538 | | 718 | |
| | (1a) | (1b) | (2a) | (2b) | (3a) | (3b) | (4a) | (4b) |
| Treatment | 0.068**** (0.002) | 0.053**** (0.002) | 0.080**** (0.002) | 0.064**** (0.002) | 0.089**** (0.002) | 0.076**** (0.002) | 0.092**** (0.002) | 0.081**** (0.002) |
| Treatment×LowP | -0.018**** (0.002) | | -0.013**** (0.002) | | -0.012**** (0.002) | | -0.007** (0.002) | |
| Treatment×LowC | | 0.011**** (0.002) | | 0.019**** (0.002) | | 0.013**** (0.002) | | 0.015**** (0.002) |
| Observations | 2,957,138 | | 2,957,749 | | 2,957,291 | | 2,957,886 | |

Note: To preserve data confidentiality, we normalize the dependent variable. Robust standard errors are reported in parentheses. $^{**}p < 0.01; ^{****}p < 0.0001$.

form specified below with polynomial degree $K = 3$:

$$\mathbb{E}[Y|\boldsymbol{X} = \boldsymbol{x}, T = t] = G(\boldsymbol{\theta}^*(\boldsymbol{x}), t) \tag{33}$$

$$= \frac{1}{1 + \exp(-(\theta_0^*(\boldsymbol{x}) + \theta_1^*(\boldsymbol{x})t + \theta_2^*(\boldsymbol{x})t^2 + \theta_3^*(\boldsymbol{x})t^3))}. \tag{34}$$

To implement this DGP, we use a structured Deep Neural Network (DNN) illustrated in Figure 2. The network takes 77-dimensional covariates $\boldsymbol{X}$ and processes them through three fully connected layers (128, 32, and 8 nodes) with ReLU activation. The output determines the nuisance parameters $\theta_k(\boldsymbol{x})$, which are then combined with the treatment $t$ to predict the outcome.

The model is trained using the Adam optimizer (Kingma & Ba, 2014) with a binary cross-entropy loss function. We set the learning rate to 0.0001 and batch size to 128. We reserve 10% of the training data as a validation set and train for 100 epochs with early stopping (patience = 5).

**Stage 2: Value Estimation.** We estimate the policy value using Neyman Orthogonal score functions. For a policy $\pi$, the estimator is:

$$\hat{V}_K^{\text{DLPT}}(\pi) = \frac{1}{|\mathcal{S}_{\text{inference}}|} \sum_{i \in \mathcal{S}_{\text{inference}}} \psi_V(\boldsymbol{z}_i, \pi; \hat{\boldsymbol{\theta}}, \boldsymbol{\Lambda})$$

$$\text{where } \psi_V(\boldsymbol{z}_i, \pi; \hat{\boldsymbol{\theta}}, \boldsymbol{\Lambda}) = H_V(\hat{\boldsymbol{\theta}}(\boldsymbol{x}_i), \pi(\boldsymbol{x}_i))$$

$$- H_{V\boldsymbol{\theta}}(\hat{\boldsymbol{\theta}}(\boldsymbol{x}_i), \pi(\boldsymbol{x}_i))\boldsymbol{\Lambda}(\boldsymbol{x}_i)^{-1}\ell_{\boldsymbol{\theta}}(y, \check{t}_i, \hat{\boldsymbol{\theta}}(\boldsymbol{x}_i)).$$

Here, the policy value function is defined as $V(\pi) = \mathbb{E}[wY - c\pi(\boldsymbol{X})]$, where $w = 0.5$ and $c = 0.1$ are platform-specified parameters representing the revenue generated from user uploads and the cost of points, respectively.

**Stage 3: Policy Learning.** We derive optimal policies by solving the ERM problem $\hat{\pi}_K^{DLPT} = \inf_{\pi \in \Pi} \{-\hat{V}_K^{\text{DLPT}}(\pi)\}$.

**Discrete policy class:** For a discrete treatment space, finding the optimal policy requires a simple empirical maximization. It involves a simple argmax over the $m$ discrete arms:

$$\hat{\pi}(x_i) = \arg\max_{t \in \mathcal{T}} \hat{\psi}(x_i, t).$$

**Continuous policy class (DNN):** When the policy is parameterized by a Deep Neural Network (DNN) with weights $w$, the estimator acts directly as a differentiable loss function. The empirical objective to be minimized can be expressed as:

$$\mathcal{L}(w) = -\frac{1}{n} \sum_i \hat{\psi}(x_i, \pi_w(x_i)).$$

Since the estimator $\hat{\psi}$ is smooth in the treatment variable $t$, gradients with respect to the network weights $w$ are computed directly via standard automatic differentiation (e.g., PyTorch `autograd`) and trained with Stochastic Gradient Descent (SGD).

### E.2. Benchmark Methods Implementation

All benchmarks use the same training dataset $\mathcal{S}_{\text{train}}$ and inference dataset $\mathcal{S}_{\text{inference}}$ as DLPT.

**Linear Regression (LR).** Assumes $y_i = \boldsymbol{\alpha}' \boldsymbol{x_i} + \beta t_i + \mu_i$. We estimate parameters on $\mathcal{S}_{\text{train}}$ and predict outcomes on $\mathcal{S}_{\text{inference}}$ to calculate ATEs and identify optimal policies.

**Logistic Regression (LogR).** Assumes $\mathbb{E}[y_i] = \sigma(\boldsymbol{\alpha}' \boldsymbol{x_i} + \beta t_i)$, where $\sigma$ is the sigmoid function. This captures diminishing returns but assumes a restrictive functional form for heterogeneity.

**Logit Choice Model (LCM).** Assumes user utility $u_i = \boldsymbol{\alpha}' \boldsymbol{x_i} + \boldsymbol{\beta} \boldsymbol{x_i} t_i + \epsilon_i$. We fit the model using Maximum Likelihood Estimation (MLE) with the "l-bfgs-b" algorithm. This allows for linear interaction effects between covariates and treatment.

**Pure Deep Learning (PDL).** Approximates $\mathbb{E}[y_i] = f(\boldsymbol{x_i}, t_i)$ using a standard DNN architecture similar to Figure 2, but without the structured nuisance parameter layer. Inputs $\boldsymbol{x_i}$ and $t_i$ pass through hidden layers directly to the output. We use the same hyperparameters as DLPT (learning rate = 0.0001, batch size = 128). While flexible, PDL lacks the bias-correction mechanism of DLPT.

**Generalized Random Forest (GRF) T-learner.** This method trains a separate random forest for each discrete experimental arm to estimate the conditional average outcome based on user covariates $X$. For policy learning, it identifies the arm with the highest predicted outcome for each user. While non-parametric, it fails to capture the continuous structural relationship between treatment levels.

**Non-personalized Cubic Spline (Spline).** This baseline acts as a global interpolation method. It operates by: (i) estimating the average outcome at each discrete experimental arm; (ii) fitting a cubic spline to these averages to find the continuous treatment level that maximizes the outcome over the full treatment support; and (iii) assigning this exact same treatment value to all users. Crucially, while this spline can recommend any continuous treatment within the range, it relies purely on population-level smoothing and does not personalize based on user covariates $X$.

**Uniform Policy (UNI).** Assigns a single non-personalized policy to all users by selecting the treatment $\pi$ that maximizes the empirical average value $\hat{V}_{\text{UNI}}(\pi)$ on the training set.

## F. Additional Empirical Results

### F.1. Detailed ATE Estimation Results

Table 9 provides the detailed breakdown of ATE estimation performance for each treatment level. DLPT achieves consistently low error rates. Note that both deep learning methods (DLPT and PDL) show slightly higher errors at boundary levels ($t = 178, 718$) compared to intermediate levels, suggesting that extrapolation at boundaries is inherently more challenging.

Table 9: Detailed Comparison of ATE Estimators Across Methods

| Level | Truth | LR Est. | LR APE | LogR Est. | LogR APE | LCM Est. | LCM APE | PDL Est. | PDL APE | DLPT Est. | DLPT APE |
|---|---|---|---|---|---|---|---|---|---|---|---|
| 178 | 0.0586 | 0.0225 | 61.63% | 0.0214 | 63.51% | 0.0214 | 63.50% | 0.0492 | 15.99% | 0.0603 | 3.00% |
| 358 | 0.0729 | 0.0402 | 44.91% | 0.0446 | 38.85% | 0.0342 | 53.04% | 0.0707 | 3.01% | 0.0719 | 1.40% |
| 538 | 0.0824 | 0.0596 | 27.73% | 0.0533 | 35.37% | 0.0595 | 27.80% | 0.0795 | 3.51% | 0.0813 | 1.32% |
| 718 | 0.0886 | 0.1051 | 18.66% | 0.1035 | 16.77% | 0.1152 | 30.02% | 0.0831 | 6.22% | 0.0857 | 3.30% |
| MAPE | | | 38.23% | | 38.60% | | 43.59% | | 7.18% | | **2.25%** |

## F.2. Robustness Check: Sample Size Scaling

We empirically validate the regret bound derived in Theorem 3.7. Figure 3 plots the policy regret of DLPT as a function of sample size $n$. The observed decline aligns linearly with $1/\sqrt{n}$, confirming our theoretical consistency rate.

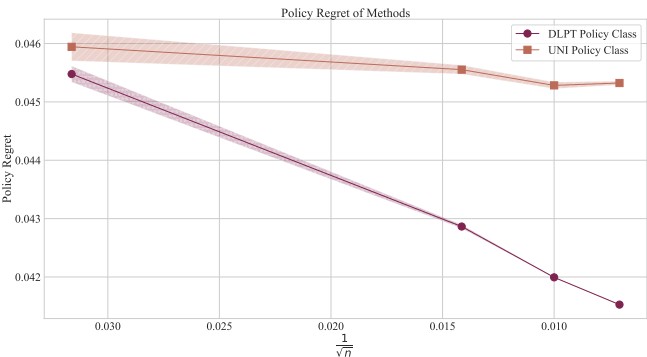

Figure 3: Policy Regret Scales with $1/\sqrt{n}$

