# OpenReview forum: "Personalized Policy Learning through Discrete Experimentation"
_ICML.cc/2026/Conference — ICML 2026 regular_

### Official Review · Reviewer_qebf · 2026-03-01

**Soundness:** 3
**Presentation:** 2
**Significance:** 2
**Originality:** 3
**Overall Recommendation:** 4
**Confidence:** 3

**Summary:**

This paper proposed the DLPT framework to learn personalized continuous policies from discrete RCTs with high-dimensional returns. The proposed estimator is consistent with mimicx optimal regret rate of $\sqrt{n}$. The proposed framework is evaluated through a large-scale experiment on a social media platform.

**Compliance With Llm Reviewing Policy:**

Affirmed.

**Final Justification:**

The rebuttal in helpful in clarifying certain points in the original manuscript. I recommend weak accept.

**Key Questions For Authors:**

1. Problem Setup: The authors assume only $m$ discrete treatment levels can be implemented in the experiments due to operational constraints. Can author provide some example of this constraints in practical applications, and is it possible to introduce a small random noise at each treatment level to smooth the observed action space?
2. Eq 2: Could author clarify do we assume the true reward follow the sigmoid-polynomial specification, or is it a user-defined parameterization to approximate the true reward?
3. Stage 3: I was wondering how do you solve the optimization problem in Eq 17? Do you consider some kind of policy gradient algorithms or other methods is considered?
4. Following my previous question, the proposed framework essentially develops a policy evaluation estimator and conducted policy optimization based on the proposed estimator. I was wondering what are the pros and cons of such method comparing directly conducted policy optimization based on the learned reward after Stage 1? Are there any considerations both from theoretical and empirical perspective?
5. Experiments: Two parameters $c,w$ are introduced in Eq 9-10, how are these two parameters determined in the experiment? Are they pre-specified?
6. The experiment essentially considers a setting where the reward is monotonic increasing with action, I was wondering if the proposed framework could handle scenarios where the reward are not necessarily monotone or even multi-modal? Will these kind of settings influence the empirical performance of the proposed framework?

Minor Comments
1. The last sentence is repeated in abstract.
2. Line 135: (See Section ?? for detailed empirical setting…).

**Limitations:**

Yes.

**Strengths And Weaknesses:**

I think this paper is trying to tackle a very interesting problem in conducting policy learning on continuous action space with discrete experimentation. The theoretical analysis and framework development are very solid and insightful. However, there are certain points in the framework (eg. the approach of policy evaluation and policy learning vs direct policy learning) that requires further discussion. Additionally, the experiment is only conducted on one setting where the reward structure is straightforward, evaluation on other settings is needed for a more comprehensive evaluation on the robustness of the proposed framework.

---

> ### Author Rebuttal · Authors · 2026-03-30
>
> **Q1: Operational constraints in practice and adding noise around treatment levels.**
> * **Statistical Power:** Concentrating traffic on a few arms is essential to ensure reliable estimates for Stage 1. Distributing users over a continuous range would lead to excessive variance.
> * **Implementation Limits:** Many real-world treatments are inherently discrete. Many treatments are inherently discrete: financial incentives use fixed platform points (preventing fractional awards like 2.4), while pricing tiers and notification frequencies (e.g., 1, 3, or 7 times/week) follow pre-set intervals
> * **Adding noise:** Dithering $T_i \sim \text{Uniform}[t_k - \delta, t_k + \delta]$ provides local gradients but cannot resolve interpolation gaps between distant arms (e.g., $t=358$ and $t=538$), and thus fails to recover the full continuous treatment curve, which is the building block of continuous policy optimization. In contrast, our framework leverages the polynomial structure of treatment to extrapolate across the full range. Beyond this theoretical limitation, dithering is also operationally difficult to implement in practice.
>
> **Q2: Does Eq. 2 assume the true reward follows the sigmoid-polynomial specification?**
>
> Equation (2) plays a dual role:
> * **Theoretically:** We assume it is correctly specified (true outcome follows $G(u(x,t;\theta^{\ast}))$) for our identification policy evaluation and optimization results.
> * **Practically:** The true DGP may deviate, so we establish an approximation guarantee: for any $s_t$-Hölder continuous $f$, there exists $\theta$ such that $\sup_{x,t} |G(u(x,t;\theta)) - f(x,t)| < o(K^{-s_t})$, where $K$ is the polynomial degree. This justifies Eq. (2) as a flexible approximation.
>
> **Q3: How is the optimization in Eq. 17 solved? Are policy gradient algorithms considered?**
>
> We will add implementation details to the appendix. No RL or policy gradient algorithms are needed:
> * **Discrete policy class:** A simple argmax over the $m$ discrete arms: $\hat{\pi}(x_i) = \arg\max_{t \in \mathcal{T}} \hat{\psi}(x_i, t)$.
> * **Continuous policy class (DNN):** The estimator acts directly as a differentiable loss: $\mathcal{L}(w) = -\frac{1}{n}\sum_i \hat{\psi}(x_i, \pi_w(x_i))$. Since $\hat{\psi}$ is smooth in $t$, gradients w.r.t $w$ are computed via standard PyTorch autograd and trained with SGD.
>
> **Q4: Pros and cons of the proposed framework vs. direct Stage 1 policy optimization.**
> * **Theoretical:** Direct Stage 1 optimization inherits nuisance estimator bias at first order. Stage 2 Neyman-orthogonal correction reduces this bias to second order ($\mathbb{E}[\nabla_\theta \hat{\psi}] = 0$). This allows us to achieve a faster rate for policy learning even if Stage 1 converges at the slower nonparametric rate ($n^{-1/4}$).
> * **Empirical:** Directly using the Stage 1 DNN-cubic model achieves 6.25% MPR. Our PDL baseline (no doubly robust correction) achieves 6.32% MPR. The full DLPT pipeline (with Stage 2 correction) achieves 0.48% MPR, confirming debiasing drives performance.
>
> **Q5: How are parameters $c$ and $w$ determined in the experiment?**
> They are pre-specified platform inputs, not estimated from experimental data: $w$ is unit profit from user engagement (a video upload), derived from downstream value like ad revenue and retention. $c$ is the exact unit cost of the incentive (e.g., 100 Points $\approx$ 0.01 USD). Together,  $H = G(w   - c \cdot \pi(x))$ captures net expected profit, reflecting standard ROI calculations.
>
> **Q6: Can the framework handle non-monotone or multi-modal rewards?**
>
>  To provide a precise answer, we would like to clarify which function is being referred to before elaborating.
>
> * **If the question concerns the net reward ($H$):**
> The net reward $H = G(\theta^{\ast}(x), t) (w - c \cdot \pi(x))$ is already non-monotone by design. Even when the outcome probability $G$ is monotonically increasing, the linear cost term $-c \cdot \pi(x)$ ensures the net reward is hump-shaped, with an interior optimum. This is precisely the structure that motivates personalized continuous policy learning.
> * **If the question concerns the outcome curve ($G$) itself:** DLPT can naturally accommodate non-monotone and even multi-modal shapes. Specifically, the cubic polynomial inside the sigmoid can inherently represent non-monotone responses (e.g., content fatigue at high incentive levels) depending on the signs and magnitudes of the learned coefficients.
> * **For highly multi-modal settings:** As established by our approximation guarantee (discussed in Q2), the framework can flexibly fit more complex, multi-modal shapes simply by increasing the polynomial degree $K$. In our specific application, the observed ATEs are monotone increasing (Table 1), which reflects the particular empirical setting but does not restrict the generality of the framework.
>
> **Q7: Minor Comments**
> We removed the redundant abstract sentence and fixed the cross-reference in Line 135.

---

> > ### Author Rebuttal · Reviewer_qebf · 2026-04-03
> >
> > Thank you for the detailed response, which addressed my major concerns. I'm happy to increase my score to weak accept.

---

> > > ### Author Response · Authors · 2026-04-04
> > >
> > > We sincerely thank you for your continued engagement and for updating the score to a weak accept. We are very glad that our detailed response successfully resolved your major concerns. We deeply appreciate the time, effort, and constructive feedback you have dedicated to evaluating and improving our work.

---

### Official Review · Reviewer_HZDd · 2026-03-07

**Soundness:** 2
**Presentation:** 2
**Significance:** 3
**Originality:** 3
**Overall Recommendation:** 4
**Confidence:** 3

**Summary:**

This paper investigates the structural limitation of standard randomized controlled trials (RCTs), which typically rely on discrete treatment arms and fail to effectively extrapolate to continuous variables or account for high-dimensional user heterogeneity. The authors seek to study a central concept of learning personalized, continuous treatment policies using only data from these discrete experiments. The article proceeds to focus on a core challenge of accurately estimating causal effects across a continuous space without suffering from nuisance estimation biases. To address this, the authors propose Deep Learning for Policy Targeting (DLPT), an innovative three-stage framework that uses a structured deep neural network to approximate the semi-parametric data generation process and applies Neyman orthogonal scores to ensure robust, unbiased policy value estimation. They validate their method theoretically by proving the estimator is asymptotically unbiased and the learned policy achieves a minimax optimal $\sqrt{n}$ regret bound , and empirically through a large-scale field experiment with over 7.3 million users on a video-sharing platform, demonstrating that DLPT significantly reduces both Mean Absolute Percentage Error and policy regret compared to existing baselines.

**Compliance With Llm Reviewing Policy:**

Affirmed.

**Final Justification:**

This paper addresses the highly practical problem of learning personalized continuous treatments using only data from discrete randomized controlled trials. My initial concerns primarily revolved around the lack of comparisons with simpler interpolation techniques and existing continuous-treatment causal inference methods. In their rebuttal, the authors provided a thorough and convincing response. They introduced a non-personalized cubic spline baseline, which effectively demonstrated the performance gains achieved by DLPT's covariate-driven personalization. Furthermore, their explanation regarding the inapplicability of existing continuous OPE methods—due to the sparse, discrete-arm nature of the experimental data—was well-reasoned. I thank the authors for addressing every single question I raised, including the minor cross-reference issues. The rebuttal has fully resolved my concerns, and I am happy to maintain my positive score of 4.

**Key Questions For Authors:**

Please check the Weaknesses.

**Limitations:**

yes

**Strengths And Weaknesses:**

**Strengths:**

1. The paper addresses a highly valuable and practical problem in offline policy learning: estimating continuous policy values and optimizing personalized continuous policies using only data collected from discrete randomized controlled trials (A/B tests). This framework allows platforms to optimize continuous decision variables without the need to modify existing discrete experimentation infrastructure.

2. The proposed DLPT framework is supported by rigorous theoretical analysis. The authors prove that their policy value estimator, which utilizes Neyman orthogonal scores, achieves asymptotic normality. Furthermore, they successfully establish that the learned policy achieves a minimax optimal $\sqrt{n}$ regret bound.

3. The authors evaluate the DLPT framework using data from a field experiment conducted on a short-video platform , assessing the method's performance in both Average Treatment Effect recovery and policy learning tasks against standard baselines.

**Weaknesses:**

1. The proposed DLPT framework is well-motivated. However, out of curiosity, I was wondering if the authors had considered comparing it against simpler interpolation or continuous smoothing techniques.

2. I noticed that the baselines evaluated in the experiments (such as Linear Regression, Logistic Regression, and pure DNNs) appear to be relatively standard predictive or discrete choice models. I am curious why the authors did not include comparisons with existing causal inference or off-policy evaluation methods specifically designed for continuous treatments.

3. There are a few unresolved cross-references in the manuscript that should be corrected, such as the placeholder "Section ??" on lines 134-135.

---

> ### Author Rebuttal · Authors · 2026-03-30
>
> **Q1: Comparison against simpler interpolation or continuous smoothing techniques**
>
> We thank the reviewer for this suggestion. We have added a baseline policy based on non-personalized cubic splines, which operates as follows:  (i) estimates $\mathbb{E}[Y \mid T=t]$ at each discrete arm; (ii) fits a cubic spline to find the continuous $t^{\*}$ maximizing $H(t)$ over the full treatment support; and (iii) assigns this same $t^{\*}$ to all users.
> Crucially, this spline baseline is not restricted to the five tested arms; it can recommend any continuous treatment within the range. If the policy were restricted to tested arms, the spline would simply select the single best-performing discrete arm for everyone, identical to our **AVG** baseline. Results on the out-of-sample validation set are as follows:
>
>
> | Method | MPR (%) |
> | :--- | :--- |
> | **DLPT (ours)** | **19.69** |
> | PDL | 29.28 |
> | Spline (non-personalized) | 30.06 |
> | AVG (best discrete arm) | 32.60 |
>
>
> If user covariates $X$ were absent, global interpolation via splines would be sufficient to find the single best treatment for the population. However, the core of DLPT is modeling $\theta^{\*}(x)$ to capture how the outcome curve varies across users. DLPT achieves 10.4 pp lower regret than the spline (19.69% vs. 30.06%) because it leverages high-dimensional covariates to personalize the optimal continuous treatment, rather than assigning the same $t^{\*}$ to the entire population.
>
>
> **Q2: Comparisons with existing causal inference or off-policy evaluation methods**
>
> We thank the reviewer for this important observation. We address this in two parts.
>
> **Causal baselines are included.** Our SDL baseline is precisely a causal off-policy method: it uses the Stage 1 doubly robust nuisance estimates to compute $\hat{\psi}(x, t)$ for each discrete arm and selects $\hat{\pi}(x) = \arg\max_t \hat{\psi}(x, t)$. This directly corresponds to the augmented IPW policy learner of Athey and Wager (2021) applied to our discrete arm setting, and serves as the main causal ablation against which DLPT is compared.
>
> **Why existing continuous-treatment methods are not directly applicable.** Methods specifically designed for continuous treatments such as Kallus and Zhou (2018) and Chernozhukov et al. (2019) assume either dense observational support over a continuous treatment domain or kernel smoothing over a rich treatment distribution. In our setting, treatment support consists of exactly five discrete arms from a randomized experiment. Applying kernel-based continuous-treatment estimators in this regime would require strong interpolation assumptions across wide gaps in the treatment space, which is precisely the challenge DLPT is designed to address through the structured polynomial model. In this sense, our framework can be viewed as a structured adaptation of these methods to the sparse discrete-arm RCT setting.
>
> We agree that a more explicit discussion of this relationship would strengthen the paper, and will add a paragraph comparing DLPT to Kallus and Zhou (2018) and Chernozhukov et al. (2019) in the related work section of the revision.
>
> **Q3: Unresolved cross-references**
>
> We thank the reviewer for pointing this out. We have identified the unresolved cross-references (e.g., in Sections 3.2 and 4.1) and will ensure all links and citations are correctly mapped in the revised manuscript.

---

> > ### Author Rebuttal · Reviewer_HZDd · 2026-04-01
> >
> > Thanks for the detailed response. My questions are resolved, and I retain my current score.

---

> > > ### Author Response · Authors · 2026-04-01
> > >
> > > Thank you for your careful evaluation of our work and for the thoughtful questions raised during the review process. We are very glad that our detailed response addressed your concerns. Thank you for your time and contribution to improving our manuscript.

---

### Official Review · Reviewer_sCGM · 2026-03-13

**Soundness:** 3
**Presentation:** 4
**Significance:** 3
**Originality:** 3
**Overall Recommendation:** 4
**Confidence:** 3

**Summary:**

This paper studies the problem of personalized policy learning in A/B testing, which tries to learn continuous treatment effects from discrete experimentation, while also accounting for user-level heterogeneity. Based on DLPT, the authors proposed a policy learning framework and established theoretical properties such as consistency and square root regret, and the method is evaluated on a real social media platform for optimizing creator incentives.

**Compliance With Llm Reviewing Policy:**

Affirmed.

**Final Justification:**

I am happy with the rebuttal and maintain my positive score.

**Key Questions For Authors:**

1. I noticed that the last sentence actually got repeated twice in the abstract. I guess it is a small typo.

2. In (2) the authors introduced a sigmoid-polynomial semiparametric policy model, where the user heterogeneity part is learned from DNNs. This raises a small question of why the treatment part is not treated symmetrically, say also using a DNN to fit? A closely related work of such factorization is actually proposed by this paper: _Causal Effect Inference for Structured Treatments_. In this work, the authors prove some guarantees that such factorization can approximate a large enough function class with an increasing number of latent dimensions.

3. For the policy learning part, I can see that it purely depends on the estimated DLPT model; in practice, do we need to incorporate certain level of exploration and regularization in the policy to achieve the regret bound?

4. For the experimental evaluation, I was expecting a larger pool of treatment values, but it seems only five values are tested. This seems to be quite a limited support set as in general it requires a more uniform and widespread set of test points to fit a good continuous model with polynomial bases. Could the authors add more discussion on how large of a support is usually required for estimating a good treatment curve?

5. How does the theory suggest the DL model enable a larger/more flexible function class? The rate of convergence for the DL model seems reminiscent of the usual nonparametric rate for Holder classes/Sobolev spaces. Plus, seems the theory mostly works for shallow and moderate-sized networks, because the total parameters is around the scale of $HL = o(n)$. Under this moderate-dimensional regime, does deep learning win over other ML algorithms such as XG-boost or random forest for fitting the nuisance functions?

**Limitations:**

Yes

**Strengths And Weaknesses:**

Strength:

1. I can see the authors are studying a very real problem. On one hand, many A/B tests are running in parallel every day in tech companies, yet nearly all of them are discrete comparisons. Therefore, it is a very natural question to ask how to target continuous treatments using the large pool of discrete tests. Intuitively, it makes sense to pool information across experiments and construct a curve/surface for continuous treatment levels.

2. The real-world experiments are also pretty strong evidences compare with synthetic evals. This is a great proof of concept that connects theory with real-world practices.

Weakness: I have some critiques on weaknesses regarding assumptions, the details of the implementation, and theoretical analysis. I will break these down into details in the incoming key question section.

---

> ### Author Rebuttal · Authors · 2026-03-30
>
> We sincerely thank the reviewer for the thoughtful feedback and literature connection.
>
> **Q1: Typo in the abstract.**
> We thank the reviewer for catching this oversight; the redundant sentence has been removed.
>
> **Q2: Connection to Kaddour et al. (2021) and symmetric DNN treatment.**
> We appreciate the connection to Kaddour et al. Their factorization is elegant and closely related. Our approach shares this spirit but makes a stronger structural commitment on the treatment side, motivated by our continuous policy learning objective.
> * **Why parametric treatment?** Our goal is policy evaluation, which necessarily requires reliable evaluation of continuous treatments beyond those observed.  A joint DNN on $(X, T)$ fails to generalize to unseen treatments. The polynomial structure introduces an economic prior (e.g., smoothness, diminishing returns) guiding extrapolation and providing a closed-form expression for any $t$, which is the building block for Stage 3 performing gradient-based optimization.
> * **Relation to Kaddour et al.:** Their approximation guarantees require increasing latent dimensionality, which could reintroduce the extrapolation problem. In contrast, DLPT's fixed polynomial features are differentiable and admit analytic gradients (which are used in our Stage 2 Neyman-orthogonal correction).
>
> **Q3: Does the policy require exploration and regularization?**
> * **Exploration:** DLPT is a purely offline method, which learns policies from perfectly overlapped RCT data. We thus do not incorporate exploration as is typical in online learning.
> * **Regularization (Complexity):** Policy class complexity (measured by its covering number) is implicitly regularized by the architecture and early stopping. As detailed in Proposition 3.6, the regret bound depends on the covering number $\log \mathcal{N}(\Pi, \varepsilon)$. For a DNN, this is $O(W^2 L \log(WL/\varepsilon))$, meaning the bound is controlled by the architecture rather than the ambient treatment space.
> * **Regularization (Robustness):** Extension of our ERM-based method to variance penalization for distributional robustness is a meaningful direction for future work.
>
> **Q4: Is a support set of five treatment values sufficient?**
> We address this practical point through identification and the bias-variance tradeoff:
> * **Identification & Validation:** A degree-$K$ polynomial is identified with at least $K+1$ distinct treatments. With our cubic model ($K=3$), four arms are mathematically sufficient. In our five-arm design, we deliberately hold out one arm for Stage 2 validation,  which tests whether the recovered curve generalizes reliably beyond the support of training data.
> * **Choice of $K$:** Increasing $K$ reduces approximation bias (error is $O(K^{-s_t})$) but increases estimation variance, as more coefficients are learned from the same $m$ arms. For typical incentive responses (smooth with diminishing returns), $K=3$ strikes a pragmatic balance, satisfying identification with one arm to spare for validation.
>
> **Q5: Convergence rates and Deep Learning vs. Tree-based methods**
> We fully agree with the reviewer's sharp observation. We clarify the nuances and provide new empirical evidence below:
>
> * **The $n^{-1/4}$ rate is a general requirement:** This MSE rate is the key requirement for Stage 1 nuisance estimation to ensure $\sqrt{n}$-consistency of the policy value estimator (Chernozhukov et al., 2018). Tree-based methods can achieve this under certain conditions and are fully compatible as drop-in replacements for Stage 1.
> * **Why DL is necessary for Stage 3:** While tree models suit Stage 1, they are structurally ill-suited for Stage 3 continuous policy optimization. By partitioning space into orthogonal regions, tree models produce piecewise constant response surfaces. Consequently, they yield zero or undefined gradients with respect to the continuous treatment $t$, preventing gradient-based optimization. Our polynomial-DNN factorization resolves this by being globally differentiable by construction.
> * **Hölder/Sobolev spaces and dimensionality:** Our DL model's worst-case rate indeed resembles standard nonparametric rates for Hölder or Sobolev spaces. However, achieving the $n^{-1/4}$ rate with classical estimators in our high-dimensional space ($d=77$) requires unrealistic smoothness (e.g., $s \geq 38.5$). Conversely, deep ReLU networks mitigate the curse of dimensionality by adapting to the data's *intrinsic* lower dimensionality, jointly training all $K+1$ coefficients using shared lower layers.
> * **Empirical Confirmation:** To validate this, we added a GRF T-learner benchmark. On real production data, restricting the policy to discrete tested arms via GRF yields 6.32% MPR, and a non-personalized continuous Spline yields 6.41% MPR. In contrast, DLPT's continuous personalized policy achieves 0.48% MPR. This confirms that DLPT substantially outperforms tree-based non-parametric baselines in practice.

---

> > ### Author Rebuttal · Reviewer_sCGM · 2026-04-01
> >
> > Re Q2, the discussion of extrapolation makes sense.
> >
> > Re Q3, thanks for the clarification that this is mostly for offline settings.
> >
> > Re Q4, I get the reason for choosing K as 3 to make a perfect polynomial identification. But a small caveat is that this is subject to assuming the treatment function is a polynomial form. In practice, it is also pretty common for a choice of a slightly more general basis than simple polynomials, say cubic splines, which fit more general treatment curves. So I was thinking that those larger basis classes would require more smooth/wider spread sampling of treatment values. But I understand the current pipeline is not shooting at the more general basis.
> >
> > Re Q5, thanks for the clarification and for the additional empirical comparisons. I understand the gap between how rigid DL theory usually is compared to the actual performance DL architectures can achieve, so I've always held these doubts regarding the broad applicability of DL-related estimation bounds. But that does not really impact my eval for how well the neural nets work for the current setting.
> >
> > Overall, I am happy with the rebuttal. Good luck with the submission.

---

> > > ### Author Response · Authors · 2026-04-04
> > >
> > > We sincerely thank you for reading our rebuttal and for the encouraging feedback.
> > >
> > > Regarding Q4, we completely agree with your insight. Exploring more general basis functions like cubic splines is a highly relevant practical direction. As you rightly point out, adopting such flexible classes would indeed require denser and more widely spread sampling of treatment values. We will ensure this tradeoff between basis flexibility and sampling requirements is noted in our discussion.
> > >
> > > Regarding Q5, we share your perspective on the gap between rigid theoretical bounds and the actual empirical success of deep learning architectures. We deeply appreciate your pragmatic approach in evaluating the neural network based on its practical effectiveness in our setting.
> > >
> > > Thank you once again for your time, your highly constructive review, and your good wishes!

---

### Official Review · Reviewer_GVrs · 2026-03-15

**Soundness:** 3
**Presentation:** 3
**Significance:** 2
**Originality:** 2
**Overall Recommendation:** 3
**Confidence:** 4

**Summary:**

The paper proposes DLPT, a neural network based solution for estimating continuous treatment effects and treatment policy learning in online recommendation/pricing systems. DLPT models nuisance parameters as sigmoid polynomial and estimates it via ERM, then the policy value, defined as a combination of outcome value minus policy cost by using orthogonal score functions, and finally policy learning is done via ERM over the proposed policy value estimator. Empirical results show significant improvement on a proprietary dataset against a set of baselines.

**Compliance With Llm Reviewing Policy:**

Affirmed.

**Final Justification:**

I have to maintain my score mainly for the following reasons:
1. the authors highlighted that the work is for offline policy learning which is different from improving recommendation systems but the major experiment setting they show is indeed a subproblem of general recommendation systems (how much money incentive you should assign to a user to maximize his/her engagement vs. which content you should promote next to maximize CTR/session length). Given this task nature, it is very hard to ignore the current industry practice where people are using large transformers intensively to do generative recommendations. I do understand the resource constraints and time limit, but the baselines I asked is merely small sequential models and non-sequential NNs from older papers. But still, the authors failed to provide the results. Instead, they claim that  the [Epistemic Neural Recommendation](https://arxiv.org/pdf/2306.14834) baseline I asked, which is a neural network off-policy learner, will degrade to a cubic spline regression.

2. the actual contribution is only policy learning extension with applications to larger datasets beyond the basic framework proposed by [Farrell et al. (2020)](https://arxiv.org/pdf/2010.14694). Upon closer checking the prior paper, I find it hard to justify the claimed contributions of the current paper.

**Key Questions For Authors:**

1. Could the authors clarify in details how the proposed method different from the one presented in [Farrell et al. (2020)](https://arxiv.org/pdf/2010.14694)?
2. More evaluations with other stronger baselines like [Epistemic Neural Recommendation](https://arxiv.org/pdf/2306.14834) could be helpful and also on public datasets like [KuaiRec](https://kuairec.com).
3. Regarding the i.i.d. dataset construction, could the authors elaborate on how this assumption is satisfied and what measures are taken to counter the temporal/location/other user features heterogeneity?
4. Please fix missing refs and typos:
    - line 25, abstract: duplicate sentences
    - line 134, right side: missing refs
    - line 193, left: "Identifiability"

**Limitations:**

Please include a section of limitations and also discuss the potential impact of the work. I believe the proposed method could be used to manipulate user preferences potentially.

**Strengths And Weaknesses:**

**Strength**
1. the presentation is clear and easy to follow: the method is a three stage pipeline built upon [Farrell et al. (2020)](https://arxiv.org/pdf/2010.14694) with each step supported by proper assumptions and theoretical justifications.
2. empirical results on the proprietary dataset shows significant improvements.

**Weakness**
1. the method's novelty is limited since most of the technical contributions highly resembles the framework established by [Farrell et al. (2020)](https://arxiv.org/pdf/2010.14694).
2. the evaluation is limited in the sense that the baselines are relatively weak and no publicly available dataset is used for clearer comparison.
3. i.i.d dataset collection assumption is quite strong and the authors don't provide analysis how this is possible for the platform to conduct in scale and how they control the temporal/location/other user features heterogeneity.

---

> ### Author Rebuttal · Authors · 2026-03-30
>
> **Q1: Differences from Farrell et al. (2020)**
>
> We acknowledge that our DLPT framework instantiates the generic framework from Farrell et al for policy evaluation, which paves the way to our ultimate goal of policy learning. The two works differ fundamentally in objective and scope:
> * **Farrell et al. (2020) is an inference framework.** Their goal is to construct valid confidence intervals and hypothesis tests for heterogeneous treatment effects. There is no policy optimization involved.
> * **DLPT is a policy learning framework.** We adapt the estimation procedure proposed in Farrell et al. to construct a policy value estimator, then optimize it over a given policy class to select the policy that maximizes value. We further establish a new theoretical result providing a $\sqrt{n}$-regret guarantee. We also provide extensive empirical studies using large-scale platform production data to validate our framework.
>
> When adapting the method, we make two key modifications: (i) Under our RCT setting, the treatment assignment mechanism is known, so the debiasing matrix $\lambda(x)$ can be computed analytically rather than estimated (see Remark 3.4); and (ii) We instantiate the treatment parameterization in the generic estimator using a sigmoid-polynomial model, which enables extrapolation to treatment values beyond those directly observed.
>
> **Q2: Evaluation with Stronger Baselines and KuaiRec**
>
> We added two baselines: (1) **GRF T-learner**: a separate random forest for each experimental arm; (2) **Non-personalized Spline**: a cubic spline fit to arm-level average outcomes. On real production data, GRF achieves 6.32% MPR and Spline achieves 6.41% MPR, compared to DLPT's 0.48% MPR, confirming that DLPT substantially outperforms both.
> Regarding non-parametric methods (ENR): continuous treatment methods require dense grids. With $m=5$ arms, these reduce to the GROUP baseline. DLPT's polynomial structure is a necessity for recovering the whole treatment curve, not a restriction.
>
> **KuaiRec semi-synthetic study (7,176 users, 53 features, $N_{val}=20,000$):**
>
> We construct a semi-synthetic study using KuaiRec (Gao et al., 2022). The real component is the user covariate distribution: we sample $X$ directly from KuaiRec user features rather than from a synthetic distribution. The synthetic component is the DGP function: we generate a random DNN (100×100×100 layers) that maps each user's features to personalized parameters, defining a sigmoid-polynomial reward function $G(u(x,t))$ for each user. Observed outcomes at the five discrete arms are then drawn from this synthetic reward model. This design preserves realistic covariate structure and feature correlations from real user data. We sample $N=10{,}000$ training users and $N_{\text{val}}=20{,}000$ held-out users for validation.
>
> | Method | MPR (%) | Method | MPR (%) |
> | :--- | :--- | :--- | :--- |
> | Oracle | 0.00 | **DLPT (Stage 3)** | **10.11** |
> | LogR | 11.30 | LR | 11.66 |
> | PDL | 12.13 | SDL | 12.14 |
> | GRF T-learner | 12.61 | AVG | 14.06 |
> | GROUP | 14.10 | Spline | 15.88 |
> *Note: $MPR = (oracle - method) / oracle \times 100\%$. DLPT achieves the lowest regret.*
>
> **Q3: The i.i.d. Assumption and Heterogeneity**
> The i.i.d. assumption operates at two levels in our setting:
> * **Random assignment:** The experiment randomly assigned each of the 7.3M users to one of five arms. This ensures treatment assignment $T_i$ is independent of potential outcomes $Y_i(t)$. Temporal/geographic variations are designed to be balanced across arms in expectation and are further validated with a randomization check.
> * **Covariates $X$:** Observed heterogeneity (temporal, location, demographic) is explicitly captured in the 77-dimensional pre-treatment covariate vector $X_i$. DLPT's DNN component $\theta^*(x)$ is designed to capture heterogeneous effects along these dimensions.
> * **Short window:** During the window, each user receives treatment exactly once. This single-exposure design eliminates within-user carry-over and learning effects, making the conditional i.i.d. assumption more plausible than in repeated-treatment settings.
>
> **Q4: Limitations and Potential Impact**
>
> We will add a dedicated **Limitations** section discussing: (i) the sigmoid-polynomial parametric assumption; (ii) the requirement for discrete RCT data; (iii) the static, single-period nature of the learned policy; and (iv) SUTVA. In our deployment, SUTVA violations are minimal as the experiment involves a small fraction of total platform traffic.
> **Regarding User Preference Manipulation:** We interpret this as a concern about long-run preference drift. This is a scope limitation (point iii); our empirical validation targets the short-term, single-period setting and does not model dynamic or cumulative treatment effects.
>
> **Q5: Fixes to Refs and Typos**
> We confirm all three will be corrected: the duplicate sentence in the abstract (line 25), the missing references (line 134), and the typo "Identifiability" (line 193).

---

> > ### Author Rebuttal · Reviewer_GVrs · 2026-04-01
> >
> > Since the paper aims at improving recommendation systems, I am really concerned with the limited evaluations. In my original comment, I requested comparison against ENR, which is a neural network with injected randomness during training. I am not convinced that a cubic spline fit could be equivalent to a neural network. The current selection of the baselines (regression, random forests) are not strong nor near the industry common practice at all (not to even mention about the wide application of large transformers). Thus, I remain highly skeptical about the practical value of the proposed method.

---

> > > ### Author Response · Authors · 2026-04-04
> > >
> > > We sincerely thank you for the continued engagement and for sharing these concerns. We realize that our paper's positioning might have caused some confusion regarding our target application, and we deeply appreciate the opportunity to re-clarify our core problem setting and the practical value of our method.
> > >
> > > 1. Offline Causal Policy Learning vs. Recommendation Systems.
> > >
> > > We would like to clarify that our goal is **not** to improve general recommendation systems (such as CTR prediction using large transformers). Instead, we study a fundamentally different problem: offline causal policy learning for continuous treatments. Our goal is to learn a policy that maximizes rewards by finding the optimal continuous treatment (e.g., specific cash incentive amounts), rather than matching content to user preferences.
> > >
> > > 2. The Core Challenge and Why Pure Non-Parametric/Online Models Fail Here.
> > >
> > > Our setting is strictly offline, constrained by data from a discrete RCT with limited, finite support (e.g., exactly 5 tested arms). The core causal challenge is to generalize and recover the entire continuous treatment curve between these wide gaps. Because the setting is offline, online exploration methods with injected randomness (like ENR or bandits) are structurally inapplicable. Furthermore, pure non-parametric approaches such as standard deep learning models or large transformers that simply treat the treatment $T$ as one additional feature typically fail to generalize and extrapolate in such causal settings. To demonstrate this, we *did* include a pure deep learning benchmark (PDL) in our evaluations, which empirically underperformed precisely because it lacks the necessary structural guarantees to recover the continuous curve.
> > >
> > > 3. The Purpose of the Baselines.
> > >
> > > We completely agree with you that a cubic spline, regression, or Random Forest is not equivalent to a neural network. Our intention in including these specific baselines was not to benchmark simple models against neural networks generally. Rather, methods like Generalized Random Forests (GRF) are the standard state-of-the-art in offline causal inference. We used them to isolate and benchmark the effectiveness of our parametric polynomial specification of the treatment, demonstrating that this specific structural design is essential for continuous treatment generalization.
> > >
> > > 4. The Practical Value and Industry Adoption.
> > >
> > > Regarding the practical value, DLPT is specifically designed to be highly actionable in industry. It enables platforms to find optimal continuous treatment assignments (such as cash incentives, pricing, or discount rates) beyond the limited discrete points tested in an experiment. Crucially, it does so without requiring any changes to the existing online serving infrastructure; it simply upgrades how the offline RCT data is analyzed. Because of this lightweight, infrastructure-free nature, our method has already seen successful, wide adoption across multiple distinct applications within the company we collaborated with.
> > >
> > > If the distinction between our offline causal policy learning setting and standard online recommendation systems has not yet been sufficiently clear, we would greatly appreciate any specific feedback on which aspects of our methodology or problem formulation remain unconvincing.
> > >
> > > We sincerely thank you for your critical feedback, which will help us significantly sharpen the paper’s positioning and clarify its scope in the final version.

---

### Decision · Program_Chairs · 2026-04-30

**Decision:**

Accept (regular)

**Comment:**

This paper proposes DLPT, a method for learning personalized continuous policies from discrete RCTs, addressing limitations of existing methods that operate at discrete levels. Reviewers agreed that this paper tackles a valuable and practical problem, and that the proposed method is supported by rigorous theoretical analysis. Most reviewers leaned toward accepting the submission. The authors should incorporate the relevant clarifications and discussion from the rebuttal into the revised manuscript.